# Multiple glutathione-*S*-transferases detoxify diverse glucosinolate-based defenses of Brassicales plants in a generalist lepidopteran herbivore (*Spodoptera littoralis*)
Ruo Sun [1] ✉, Samantha Römhild [1], Yoko Nakamura[2], Michael Reichelt [1], Katrin Luck[1,2], Duc Tam Mai[1], Beate Rothe[1], Jonathan Gershenzon [1] ✉ & Daniel Giddings Vassão [1,3] ✉

Brassicales plants defend themselves with glucosinolates that, upon herbivory, are hydrolyzed into toxic isothiocyanates (ITCs) and other derivatives. The side chain diversity of glucosinolates results in a range of structurally distinct products, but how this chemical variation affects herbivores and their detoxification responses remains incompletely understood. Here, we show the effects of ITC hydrolysis products with various side chains on *Spodoptera littoralis* larvae and their detoxification system. ITCs inhibit larval growth to varying degrees, depending on the chemical nature of their side chain. The larvae metabolize ITCs by conjugating them to glutathione in the mercapturic acid pathway and to lysine forming an amine conjugate. Over half of the 34 *S. littoralis* glutathione-*S*-transferases (GSTs), tested as His-tagged derivatives, actively conjugate ITCs, with most catalyzing reactions with multiple substrates. Larval performance on various ITC-containing diets correlates positively with GST activity, highlighting this detoxification system's role in supporting growth on glucosinolate-containing plants. The propensity of multiple GSTs to react with an individual ITC and the wide expression of GST-encoding genes across larval organs likely promote the ability of this generalist herbivore to thrive on glucosinolate-defended Brassicales plants. These findings provide insight into herbivore adaptation and may inform future research on plant–insect interactions.

Plants produce an enormous range of structurally diverse metabolites to defend against herbivores[1–3]. Among these, glucosinolates (GSLs), a group of amino acid-derived natural products with approximately 100 currently elucidated structures, serve as key defensive chemicals in Brassicales plants[4]. After millions of years of evolution, the production and accumulation of different GSLs have come to vary among Brassicales plants, as well as among different plant organs, developmental stages, and even geographic locations[5–8]. This variability in GSL content may enhance the ability of Brassicales plants to adapt to different herbivores[9]. All GSLs feature an *O*-sulfated (*Z*)-thiohydroximate core connected to a β-D-glucopyranose unit and an amino acid-derived side-chain, with their structural diversity

stemming from their variable side-chain origin and modification during biosynthesis. GSLs themselves are pro-toxins stored in sulfur-rich S-cells at high concentrations until they become activated upon plant damage by spatially separated β-thioglucoside glucohydrolases (myrosinases) to produce isothiocyanates (ITCs) and other hydrolysis products[10]. Often regarded as the stereotypical dominant hydrolysis products of glucosinolates, the ITCs are toxic to many herbivores and pathogens. Their toxicity to plants, however, is minimized since ITCs are only formed at the site of damage and not in the rest of the plant[11].

ITCs have been intensively studied for their biological activities, especially in mammals, where they are considered to have anticarcinogenic

[1]Department of Biochemistry, Max Planck Institute for Chemical Ecology, Jena, Germany. [2]Department of Natural Product Biosynthesis, Max Planck Institute for Chemical Ecology, Jena, Germany. [3]Present address: Max Planck Institute of Geoanthropology, Jena, Germany. ✉e-mail: rsun@ice.mpg.de; gershenzon@ice.mpg.de; vassao@ice.mpg.de

properties[12,13]. Their highly electrophilic −N=C=S core reacts readily with many biological nucleophiles[14], through conjugation to exposed nucleophilic residues and cleavage of disulfide bonds, resulting in the disruption of protein tertiary structures and active sites[15,16]. In the plant order Brassicales, ITCs exhibit a wide variety of side-chain structures, which contribute to their diverse biological activities[17]. ITC side chain lipophilicity should facilitate diffusion through lipid bilayer membranes, allowing access to the intracellular space[18], and electron-withdrawing groups on the side chain may increase overall electrophilicity and reactivity of the –N=C=S core[19]; Thus, in addition to the properties of the –N=C=S group, the structural diversity of ITC side chains could have a large influence on their biological activities.

Faced with plant toxins, herbivorous animals have developed a large array of successful strategies to avoid the toxic effects of these compounds, enabling some species to thrive on a diverse range of host plants with apparent impunity. To overcome the GSL-myrosinase defense system of plants of the Brassicales, herbivores have evolved multiple mechanisms[20,21]. For example, some herbivores specializing on GSL-containing plants circumvent ITC formation by rapidly metabolizing GSLs into products that cannot be activated[22,23]. Herbivores with broader dietary ranges, on the other hand, often utilize general and conserved detoxification mechanisms, letting ITCs first form and then detoxifying them, for example, by conjugation to amino acids. This conjugation alters the reactivity and polarity of the ITCs and facilitates their efflux from cells[12,24]. For example, human cells can conjugate benzyl ITC with lysine (Lys) in vitro[25]. The typical detoxification reaction for generalist-feeding herbivores, however, involves the conjugation of ITCs with the reduced tripeptide glutathione (γ-Glu-Cys-Gly, GSH)[12]. The initial conjugates are then transformed sequentially to ITC-cysteinylglycine (ITC-CysGly), ITC-cysteine (ITC-Cys), and ITC-*N*-acetylcysteine conjugates (ITC-NAC) through the mercapturic acid pathway that predominates in mammals and in most generalist herbivores, including the Egyptian cotton leafworm *Spodoptera littoralis* (Lepidoptera: Noctuidae)[24,26,27]. As a major polyphagous pest in Africa, Mediterranean Europe, and Asia, *S. littoralis* infests more than 80 economically important plant species, including those in the Brassicales order[28]. When the larvae of this polyphagous pest fed on 4-methylsulfinylbutyl (4MSOB) ITC in artificial diet, the major metabolites were GSH-derived conjugates, but more than half of the ITC ingested was excreted in its original form[24,27]. On diets with high concentrations of 4MSOB ITC (2-4 μmol per g fresh weight), *S. littoralis* growth was lower. This was attributed to decreases in GSH, which led to declines in Cys and total protein[29]. Further studies on *S. littoralis* involving ITCs with different side-chain structures and lower natural ITC doses as found in typical Brassicales plants have yet to be conducted.

The enzymes that catalyze the initial conjugation of GSH with ITCs are glutathione-*S*-transferases (GSTs). This large enzyme family also catalyzes the reaction of GSH with other electrophilic substrates, including further plant defensive chemicals and insecticides[30,31]. In insects, the subfamily of cytosolic GSTs is grouped into six classes: delta (GSTD), epsilon (GSTE), zeta (GSTZ), theta (GSTT), sigma (GSTS) and omega (GSTO)[32]. Particularly, GSTs belonging to the delta and epsilon classes have been implicated in the detoxification of various plant defensive chemicals, including ITCs[31,33]. GSTE1 in *Spodoptera litura* and GSTE7, GSTD1, and GSTD2 in *Drosophila melanogaster*, for instance, have been associated with ITC detoxification[34–37]. Additionally, the transcript levels encoding *GSTE1* in *S. litura* and *GSTD2* in *D. melanogaster* can be induced by ITC treatment[34,37]. However, the involvement of GSTs from the other classes in detoxification remains largely unexplored. Comparing different ITCs, the activities of *D. melanogaster* GSTD1 and a *Spodoptera frugiperda* midgut preparation containing GSTs were found to vary with the ITC substrate[35,38]. Thus, further investigation should be carried out with a larger variety of ITCs to determine if there are any patterns of specificity contingent on the chemical structure of the side chain.

In this study, we investigated the mechanisms employed by the generalist herbivore *S. littoralis* to detoxify a diverse range of ITCs from the hydrolysis of GSLs in Brassicales plants. First, we determined the

physiological effects of different ITCs on *S. littoralis* larval growth and development. Next, using non-targeted metabolomics analyses, we aimed to identify the major products of ITC metabolism formed during feeding. After observing that all ITCs examined were substantially metabolized via conjugation to GSH and the mercapturic acid pathway, we studied the specific activities of all cytosolic *S. littoralis* GSTs as His-tagged derivatives towards ITCs with diverse side chain structures. Finally, we explored the expression patterns of GST-encoding genes across different *S. littoralis* organs. We found that this generalist-feeding herbivore appears to use many members of its large GST enzyme family to manage a chemically diverse range of toxins in its potential food plants.

## Results

### Inhibition of *Spodoptera littoralis* growth by ITCs depends on side-chain structure

Herbivory of Brassicales plants by *S. littoralis* caterpillars triggers the activation of GSLs, leading to the formation of toxic ITCs[24]. To assess the effects of ITCs with varying side chain structures on *S. littoralis* larval growth rate, larvae were fed an artificial diet supplemented with a realistic natural concentration of 1 μmol per g ITC and compared to those fed a negative control diet without ITCs and positive control diets with insecticides[39]. Larval growth was slower on all ITC-supplemented diets compared to the negative control group, including diets with ITCs containing linear or branched, saturated or unsaturated, and benzenic or aliphatic side-chains (Fig. 1A). Larval growth was most severely impacted by an aliphatic ITC with an organosulfur function, a terminal methylsulfinyl group: 4-methylsulfinylbutyl (4MSOB) ITC, which caused similar inhibition to an artificial diet with insecticides (Fig. 1A and Supplementary Fig. 1A). *S. littoralis* larvae performed better on diets containing allyl ITC, butyl ITC and *sec*-butyl ITC, compared to those with 4MSOB ITC; and they also performed better on benzyl ITC than on 2-phenylethyl (2PE) ITC (Fig. 1A). This indicates that side-chain length and chemistry affect ITC toxicity and consequently larval development.

Although supplementation of artificial diet with ITCs reduced *S. littoralis* larval growth, it did not impact survivorship (Fig. 1B), unlike the group treated with insecticides in which most of *S. littoralis* larvae did not survive after 7 days feeding (Supplementary Fig. 1B). However, in contrast to previous results, the total amounts of soluble protein and free amino acids in larvae did not decrease with ITC feeding (Fig. 1C). In fact, cysteine levels in larvae fed on 4MSOB ITC diets and lysine levels in larvae fed on 4MSOB ITC and benzyl ITC diets were even elevated compared to the control group (Fig. 1C). This suggests that *S. littoralis* can adjust to ITC-containing diets containing lower natural concentrations of these plant-derived toxins and not suffer declines in amino acid content due to ITC metabolism.

To investigate the impact of ITCs on *S. littoralis* larvae feeding on actual plants, we used *Arabidopsis thaliana* lines with varying GSL profiles: Col-0 wild-type (with aliphatic and indolic GSLs), *myb28myb29* knockout (KO) mutants (with indolic GSLs remaining), *myb28myb29* × *cyp79b2cyp79b3* KO mutants (lacking all GSLs), and *CYP79A2* knock-in (KI) mutants (with aliphatic, indolic and benzenic GSLs). 4MSOB GSL is the predominant aliphatic GSL in wild-type and *CYP79A2* KI plants, but is not present in *myb28myb29* KO and *myb28myb29* × *cyp79b2cyp79b3* KO plants, while indolic GSLs, such as indolyl-3-methyl (I3M) GSL, are present in wild-type, *myb28myb29* KO, and *CYP79A2* KI plants (Fig. 1D, Supplementary Table 1). Larvae that fed on wild-type plants exhibited significantly slower growth rates than those that fed upon *myb28myb29* × *cyp79b2cyp79b3* KO and *myb28myb29* KO plants, which lack aliphatic GSLs such as 4MSOB GSL, the precursor of 4MSOB ITC (Fig. 1E). Larval growth was even slower on *CYP79A2* KI plants (containing added benzyl ITC) compared to wild-type plants (Fig. 1D, E). Considering total GSLs, larval growth was slowest on lines with greater GSL content, although the relationship was not linear. On the other hand, I3M GSL did not appear to hinder larval development, as seen by comparing larvae fed on *myb28myb29* KO and *myb28myb29* × *cyp79b2cyp79b3* KO plants (Fig. 1E). These results indicate that both the total amount and the structure of GSLs influence larval growth,

**Fig. 1 | *Spodoptera littoralis* larval growth is inhibited by isothiocyanate (ITC) feeding, with inhibition dependent on ITC chemical structure.**
**A** Growth of *S. littoralis* larvae was negatively affected by the presence of ITCs in artificial diets ($n = 15$). Relative growth rates ($\text{mg} \cdot \text{mg}^{-1} \cdot \text{day}^{-1}$) are calculated as the natural log (Ln) of the final day's mass minus the natural log of the initial day's mass, divided by the number of feeding days (12 days for all treatments). **B** The survivorship of *S. littoralis* larvae was not significantly influenced by ITC feeding ($n = 15$). **C** The soluble protein and free amino acid contents of *S. littoralis* larvae fed on artificial diets with ITCs were not reduced and were even enhanced. Data show the $\log_2$ fold-change of amino acid concentration in ITC treatments normalized relative to the control ($n = 5$).
**D** Glucosinolate (GSL) content of *Arabidopsis thaliana* wild-type plants and *myb28myb29* KO (knock-out) mutants, *myb28-myb29 × cyp79b2cyp79b3* KO mutants, and *CYP79A2* KI (knock-in) mutants, which contain different structures and amounts of GSLs ($n = 5$ for each treatment). The GSLs included under "Other GSLs" are: 4OHI3M (4-hydroxy-3-indolylmethyl) GSL, 7MSOH (7-methylsulfinylheptyl) GSL, 4MTB (4-methylthiobutyl) GSL, and 1MOI3M (1-methoxyindol-3-ylmethyl) GSL. The detailed GSL content of these lines is listed in Supplementary Table 1.
**E** *S. littoralis* larval development was affected by feeding on these *A. thaliana* plants ($n = 15$). Relative growth rates ($\text{mg} \cdot \text{mg}^{-1} \cdot \text{day}^{-1}$) are calculated for 3 days of feeding. Abbreviations: 2PE ITC 2-phenylethyl ITC, GSH glutathione, GSSG glutathione disulfide, 8MSOO GSL 8-methylsulfinyloctyl GSL, 5MSOP GSL 5-methylsulfinylpentyl GSL, 4MSOB GSL, 4-methylsulfinylbutyl GSL, 3MSOP GSL 3-methylsulfinylpropyl GSL, 4MOI3M GSL 4-methoxyindolyl-3-methyl GSL, I3M GSL indolyl-3-methyl GSL. Box graphs indicate the range of 25–75%, the middle lines indicate the median values, and the whiskers indicate the range of data points up to 1 time the interquartile range. Significant differences between means (± SE) were determined by Tukey HSD tests in conjunction with one-way ANOVA in (**A**, **C**, and **E**), and with two-way ANOVA in (**D**). Survival analysis was determined using Kaplan–Meier survival estimates in (**B**). Different lowercase letters or asterisks in the graph denote statistically significant differences based on $P < 0.05$.

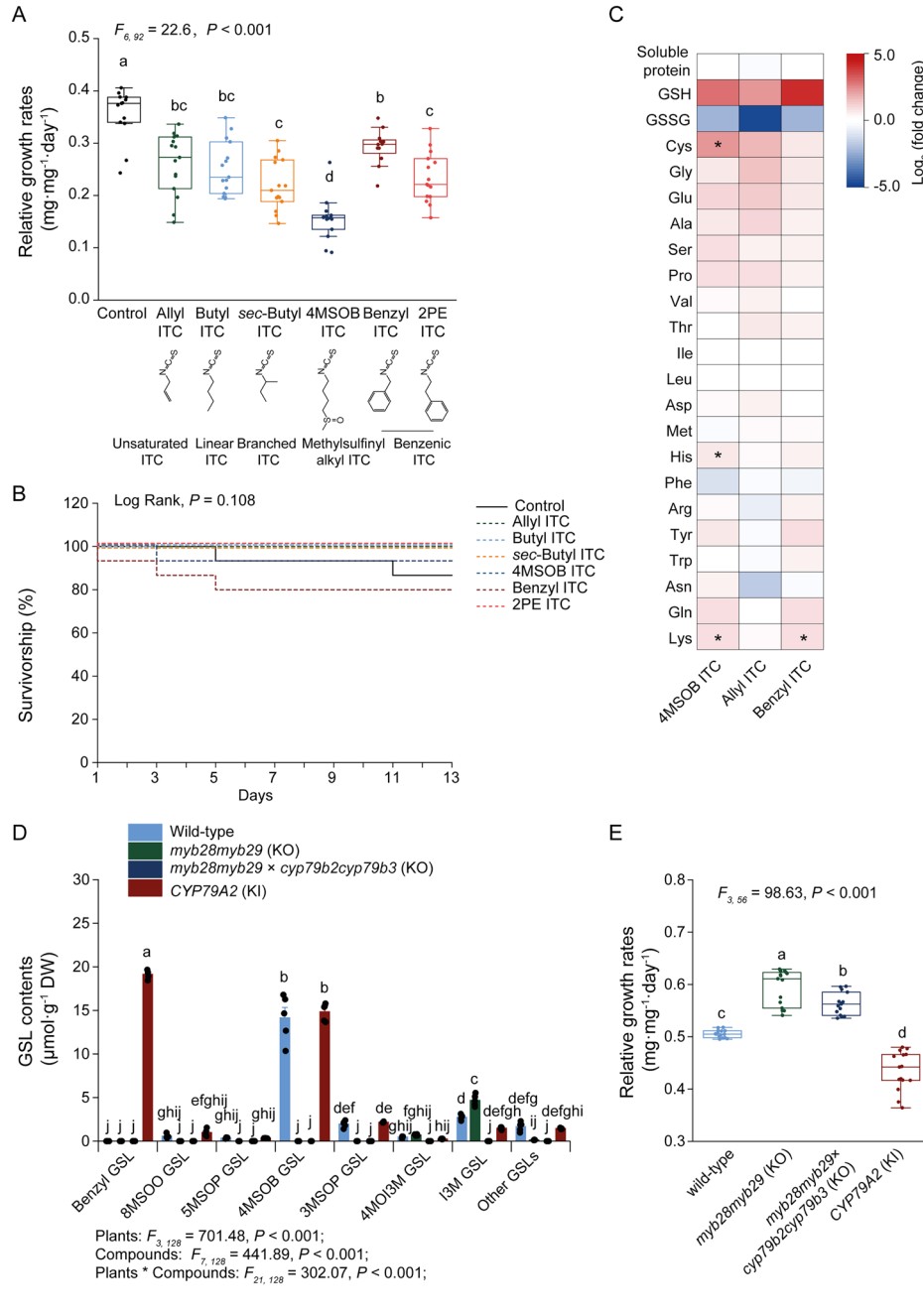

and that 4MSOB GSL was especially toxic, as also seen in the artificial diet experiments.

## *S. littoralis* larvae metabolize diverse ITCs to GSH and lysine conjugates

To determine why ITCs reduce *S. littoralis* growth, we investigated their metabolism. Non-targeted metabolomics analyses were employed using ultra-high-performance liquid chromatography coupled to quadrupole time-of-flight mass spectrometry (UHPLC-qTOF-MS) to look for glutathione conjugates and further metabolites. A number of compounds were detected in the frass of larvae fed on a diet with 4MSOB ITC, including 4MSOB ITC, 4MSOB ITC-GSH, 4MSOB ITC-CysGly, 4MSOB ITC-Cys, and 4MSOB ITC-Cyclic-Cys (Fig. 2A). Additionally, 4MTB ITC conjugates (4MTB ITC-CysGly, 4MTB ITC-Cys, and 4MTB ITC-Cyclic-Cys) were detected due to the presence of 1.85% 4MTB ITC in the commercial 4MSOB

ITC preparation[40]. These metabolites demonstrate that after ITCs are conjugated to GSH, the glutamate and glycine residues are sequentially removed, followed by intramolecular cyclization (Fig. 2B). Using targeted metabolite analyses with liquid chromatography-mass spectrometry (LC–MS/MS), we determined that the 4MSOB ITC-Cys conjugate was the predominant 4MSOB ITC derivative in the hemolymph, integument, and frass of *S. littoralis* larvae. Over 60% of the total 4MSOB ITC in larval frass was found in the form of conjugates, with more than half being 4MSOB ITC-Cys (Fig. 2C and Supplementary Fig. 2). The total conversion of 4MSOB ITC into its conjugates by *S. littoralis* was assessed by quantifying the amounts of 4MSOB ITC and its conjugates in the consumed artificial diet, larval bodies, and excreted frass. Approximately 36% of the ingested 4MSOB ITC was converted into conjugates and excreted in the frass, while 8% of the conjugated forms remained within the larval body (Supplementary Fig. 2). When fed with artificial diets with other ITCs, *S. littoralis* also

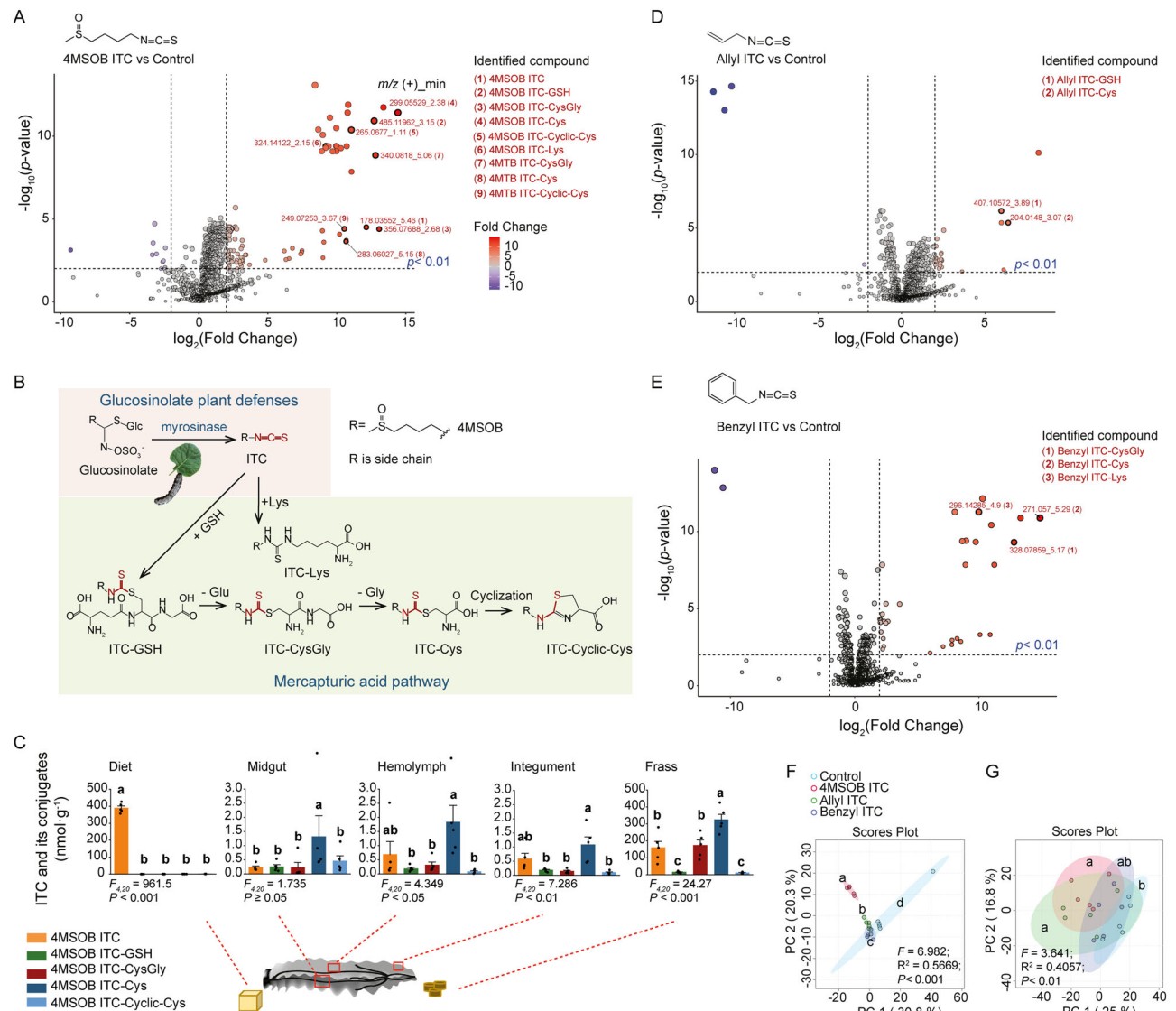

**Fig. 2 | Metabolism of ITCs to GSH conjugates and derivatives by *S. littoralis* larvae. A** Volcano plot of extracted LC-MS/MS features from non-targeted Q-TOF (UHPLC-HRMS, positive mode) analyses of frass of *S. littoralis* fed on an artificial diet supplemented with 4MSOB ITC compared to frass from feeding on an artificial diet without 4MSOB ITC (control) ($n = 5$ for each treatment). Important features were identified by co-chromatography with standards that were commercially available or produced by chemical synthesis (Supplementary Fig. 5), and are highlighted in figures using black rings around the red dots. The metabolites of 4MTB ITC, comprising 1.85% of commercial 4MSOB ITC[40], were also detected. **B** Chemical structures of ITC conjugates measured in panel A and their likely metabolic sequence. **C** Targeted analyses of 4MSOB ITC and derivatives in different tissues (midgut, hemolymph, integument, and frass) of *S. littoralis* larvae fed on an artificial diet containing 4MSOB ITC ($n = 5$). **D, E** Volcano plots of extracted LC–MS/MS features from non-targeted metabolomics analyses indicating the major

*S. littoralis* metabolites after feeding allyl ITC (**D**) and benzyl ITC (**E**) ($n = 5$). **F, G** Principal component analysis (PCA) plots showing the variation among metabolites in the frass (**F**) or bodies (**G**) of *S. littoralis* larvae fed on artificial diets containing 4MSOB ITC, allyl ITC, benzyl ITC, or without ITC (negative control). LC-MS/MS features were extracted from non-targeted Q-TOF (UHPLC-HRMS, positive mode) analyses ($n = 5$). Detailed features of volcano plots with statistically significant differences are listed in Supplementary Data 6. Detailed results of metabolomics analyses of frass and bodies of *S. littoralis* fed on artificial diets containing ITCs are listed in Supplementary Data 1 and Supplementary Data 2, respectively. Statistically significant differences between means (± SE) were determined by two sample t-tests ($P < 0.01$) in (**A, D**, and **E**), by Tukey HSD tests ($P < 0.05$) in conjunction with one-way ANOVA in (**C**), and by pair-wise PER-MANOVA ($P < 0.01$) in (**F, G**). Different lowercase letters or asterisks denote statistically significant differences.

formed ITC-GSH conjugates and their derivatives, particularly ITC-Cys conjugates, but there were differences compared to 4MSOB ITC. Specifically, allyl ITC was converted chiefly to allyl ITC-GSH and allyl ITC-Cys (Fig. 2D), while benzyl ITC was metabolized primarily to benzyl ITC-CysGly and benzyl ITC-Cys conjugates (Fig. 2E). Metabolism of 2PE ITC (Supplementary Fig. 3A) and butyl ITC (Supplementary Fig. 3B) gave other patterns. Significant metabolomic differences were observed in larval frass after ITC feeding compared to non-ITC feeding (Fig. 2F). In the larval body, significant differences appeared with 4MSOB ITC or allyl ITC feeding but not with benzyl ITC compared to non-ITC feeding (Fig. 2G). Moreover,

there was more significant variation observed in the frass than in the bodies after ITC feeding (Fig. 2F, G).

Apart from mercapturic acid pathway metabolites, ITC-lysine (Lys) conjugates were also detected in the frass of *S. littoralis* fed on 4MSOB ITC, benzyl ITC, and 2PE ITC (Fig. 2A, E, Supplementary Fig. 3A), indicating that there are other strategies for *S. littoralis* to metabolize these GSL hydrolysis products. The structures of all ITC conjugate products, including both Lys and GSH conjugates and their derivatives, were confirmed by chemical synthesis followed by mass spectrometry and NMR analyses (Supplementary Note 1 and Supplementary Fig. 4), or comparison to

commercially available standards via co-chromatography and mass spectrometry (Supplementary Fig. 5).

The GSL metabolism of *S. littoralis* larvae feeding on plants was also investigated with metabolomics analyses conducted on larvae fed on *A. thaliana* wild-type plants and the transgenic *myb28myb29* KO, *myb28-myb29 × cyp79b2cyp79b3* KO, and *CYP79A2* KI mutants. The 4MSOB ITC, 4MSOB ITC-CysGly, 4MSOB ITC-Cys, and 4MSOB ITC-Cyclic-Cys conjugates were detected in the frass of insects fed on wild-type and *CYP79A2* KI plants, with 4MSOB ITC detected in their bodies (Supplementary Fig. 6A). Meanwhile, the benzyl ITC-Cys and benzyl ITC-Cyclic-Cys conjugates were detected in the frass of *S. littoralis* fed on *CYP79A2* KI *A. thaliana* plants (Supplementary Fig. 6A). Significant metabolomic differences were found among *S. littoralis* larvae fed on different plants. Comparing different plant lines, there was more significant variation observed in the frass than in the bodies (Supplementary Fig. 6B, C).

### *S. littoralis* GST enzymes have specificities towards ITCs with different side chain structures

Since most *S. littoralis* ITC metabolites identified were derived from GSH conjugates, we investigated conjugate formation, which is catalyzed by members of the glutathione-*S*-transferase (GST) enzyme family. The specific activity of GSTs towards ITCs with varying side-chain structures, including unsaturated, linear, branched and organosulfur aliphatic ITCs, as well as benzenic ITCs, was investigated with 34 cytosolic GST proteins of *S. littoralis* (out of 37 putative GSTs, including 3 membrane-associated proteins in eicosanoid and GSH metabolism (MAPEGs))[32]. *S. littoralis* cytosolic GSTs were classified into six subfamilies, namely GSTD, GSTE, GSTZ, GSTT, GSTS, and GSTO, based on conserved domains within the amino acid sequence (Fig. 3A). Two unclassified GSTs are referred to as GSTU1 and GSTU2 (Fig. 3A). To elucidate the involvement of particular GSTs in ITC metabolism, all 34 cytosolic GST-encoding genes present in *S. littoralis* were cloned, and the corresponding enzymes expressed in *Escherichia coli* cells as His-tagged derivatives. The heterologously expressed proteins were purified, and their specific activities with ITCs of varying side-chain structures were determined. ITCs and GSH were incubated with the His-tagged GSTs, and the resulting ITC-GSH conjugates were quantified based on their absorption at 274 nm. GST enzymes from the delta, epsilon and sigma classes conjugated ITCs with GSH, whereas enzymes from the zeta, theta and omega classes largely did not (Fig. 3B, Supplementary Fig. 7). In general, the specific activities were negatively correlated with ITC side-chain length and branching (Fig. 3B, C, Supplementary Table 2). Specifically, activity was higher for allyl ITC than butyl ITC, higher for butyl ITC than isobutyl ITC and *sec*-butyl ITC, and higher for benzyl ITC than 2PE ITC (Supplementary Fig. 7). With a few exceptions (such as GSTE1 and GSTE10), GST enzymes rarely had activity on ITCs with a methylsulfinyl function, namely 3MSOP ITC and 4MSOB ITC (Fig. 3B). Most active GST enzymes exhibited high activity with benzyl ITC and the unsaturated aliphatic allyl ITC (Fig. 3B).

Principal component analysis was conducted to show the association of GST enzyme activities with different ITC structures. GSTE9 was the enzyme with the highest specific activity towards the unsaturated allyl ITC; GSTS5 and GSTS6 showed high specificities towards benzyl ITC; and GSTE14 exhibited notable specificity towards both allyl ITC and benzyl ITC. However, no GST displayed such specific targeting towards branched ITCs or those with a methylsulfinyl group (Fig. 3D). Kinetic assays of GST enzymes with ITCs further confirmed that GSTE9, GSTE14, and GSTE17, all from the epsilon class, had higher activities ($V_{max}$) and higher catalytic efficiencies ($k_{cat}/K_M$) toward allyl ITC than benzyl ITC, with GSTE9 displaying remarkably high enzyme activity for both of these ITCs (Table 1). Conversely, GSTD3, GSTS5, and GSTS6 from the delta and sigma classes exhibited higher catalytic efficiency towards benzyl ITC than allyl ITC (Table 1). The kinetic assay data confirmed that GSTs from four families are active in vitro with the same ITCs, albeit with varying efficiency, and might cooperate in vivo to metabolize these plant-derived toxins.

The variable enzymatic activities of GSTs toward different ITCs prompted us to perform molecular docking simulations to further

investigate the substrate–protein interactions with a model binding site based on crystal structures of other insect GSTs. Our goal was to gain mechanistic insight into the structural features that influence substrate recognition and binding affinity. As summarized in Table 2, benzyl ITC and 2PE ITC exhibited lower binding energies compared to allyl ITC and butyl ITC, suggesting that benzyl and 2PE ITCs bind more strongly to GSTs. Consistently, the predicted inhibition constants for benzyl ITC and 2PE ITC were also lower than those for allyl ITC and butyl ITC when interacting with GSTE9, GSTD3, GSTS5, and GSTS6 (Table 2). Molecular docking simulations further revealed that benzyl ITC forms stronger hydrogen bonding interactions with GSTE9 and GSTS5 compared to allyl ITC, as illustrated in the overlay views (Fig. 3E). These results support the conclusion that the side chain structure of ITCs significantly influences their binding affinity and inhibitory interaction with GST proteins.

### GST-encoding genes are broadly expressed in *S. littoralis* larval tissues, but only a few are inducible by dietary ITCs

To explore tissue-specific expression patterns of *S. littoralis* GST-encoding genes, we measured the transcript levels of 34 cytosolic *GST* genes across various larval tissue types, normalized relative to the expression of the ribosomal protein L13-encoding gene (*rpl13*). Transcript levels were compared in the midgut, hemolymph, integument (skin and fat body), and Malpighian tubules of *S. littoralis* larvae (Fig. 4A, B). Overall, the hemolymph showed the lowest *GST* gene expression among the tissues analyzed (Fig. 4A, B). The *GST* genes of the epsilon and delta classes were expressed across all tissues but weakly in the hemolymph (Fig. 4A, B). In contrast, genes from the zeta, theta, and omega classes exhibited broader expression across all tissues. Notably, sigma class *GST* genes showed tissue-specific patterns: *GSTS6* was predominantly expressed in the midgut, *GSTS5* and *GSTS4* were most expressed in the integument, and *GSTS2* showed higher expression in the Malpighian tubules (Fig. 4A, B).

To assess the inducibility of *GST* expression by ITC feeding, we analyzed midgut tissues from larvae fed an artificial diet supplemented with 4MSOB ITC, allyl ITC, or benzyl ITC compared to a non-ITC control diet (Supplementary Fig. 8). Of the 34 genes analyzed, only three (*GSTE6*, *GSTS3*, and *GSTS6*) were inducible, and all were specifically upregulated by benzyl ITC feeding (Fig. 4C). Following benzyl ITC feeding, *GSTE6* expression increased three-fold, while *GSTS3* and *GSTS6* expression increased two-fold compared to the control diet (Fig. 4C). The remaining *GST* genes were not induced by benzyl ITC or by allyl ITC or 4MSOB ITC. Interestingly, the expression of several *GST* genes was even down-regulated by ITC feeding (Supplementary Fig. 8).

## Discussion

The GSLs of the Brassicales are one of the best-studied groups of plant anti-herbivore and anti-pathogen defenses[17,41]. As with other defense compounds, a large number of GSL structures have been described, and these have been divided into aliphatic, benzenic and indolic classes, based on the nature of their side chains[4]. This diversity is amplified by the formation of different kinds of hydrolysis products from a single GSL, including ITCs, nitriles, oxazolidine-2-thiones and epithioalkenes[21]. ITCs are among the major toxic hydrolysis products derived from most aliphatic GSLs[11]. However, the defensive capabilities of individual ITCs have only infrequently been compared within a single herbivore. In this study, we found that ITCs with complex structures and larger side chains tended to have a stronger negative impact on *S. littoralis* larval growth than smaller, structurally simpler ITCs. For example, performance was poorer with 4MSOB ITC than with butyl ITC or *sec*-butyl ITC (Fig. 1A). In fact, 4MSOB ITC with its sulfinyl group was the most growth inhibitory of the ITCs tested, consistent with other studies[42]. We also confirmed the strong impact of 4MSOB ITC and other aliphatic ITCs in bioassays with *A. thaliana* lines with different GSL compositions. Here, feeding *S. littoralis* on lines with 4MSOB GSL and other aliphatic GSLs reduced their growth more than feeding on lines with none or only indolic GSLs (Fig. 1D, E). Other chewing herbivores, such as the lepidopterans *Manduca sexta* and *Trichoplusia ni*,

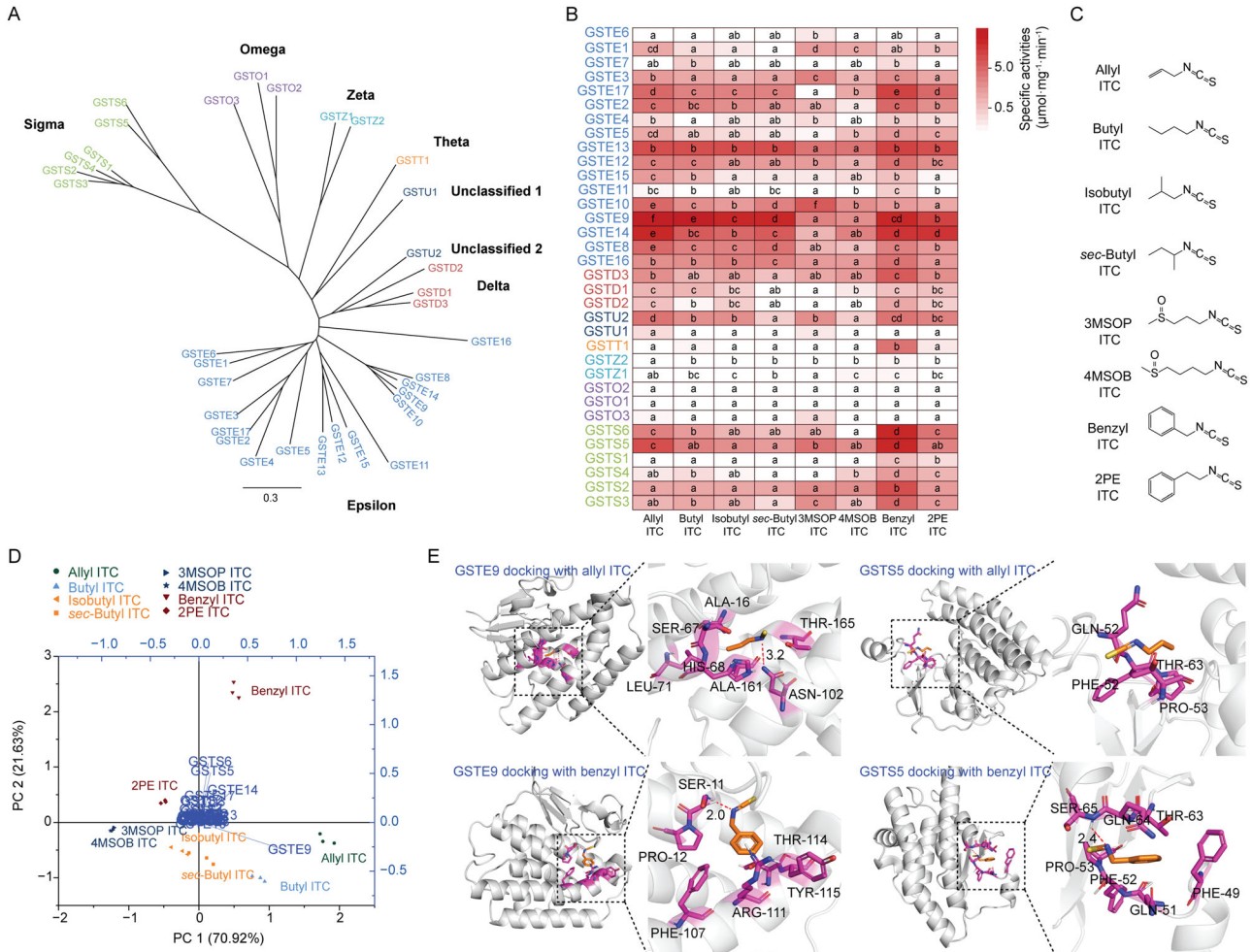

**Fig. 3 | Substrate specificity of *S. littoralis* GSH-*S*-transferases (GSTs) for ITCs with different chemical structures. A** Phylogenetic analysis of *S. littoralis* GSTs. The tree was constructed from amino acid sequences of 34 GSTs classified into six groups (the delta, epsilon, zeta, theta, sigma and omega classes) and two unclassified enzymes. The analysis was conducted using a neighbor-joining method with the Clustal Omega 1.2.2 program. The scale bar represents 0.3 expected amino acid substitutions per site. The identity% of GSTs based on amino acid sequences is listed in Supplementary Data 7. **B** The specific activity ± SE (μmol substrate consumed mg$^{-1}$ enzyme min$^{-1}$) of *S. littoralis* GSTs towards ITCs was determined by assay of the expressed and purified His-tagged proteins. Proteins were ordered by amino acid sequence identity, as shown in panel (**A**). ITCs tested include unsaturated, linear, branched and methylsulfinyl-containing aliphatic ITCs, as well as

benzenic ITCs (*n* = 3 for each treatment). Actual specific activity values corrected for non-enzymatic conjugation are given in Supplementary Table 2. **C** The chemical structure of ITCs used for GST enzyme assays. **D** A PCA plot showing the variation in the activity of *S. littoralis* GSTs with different ITCs. Vectors indicate the direction and strength of each GST protein activity relative to the overall distribution. Colored symbols correspond to the five chemical types of ITCs used in this study. **E** Molecular docking simulations of the overlay view of GSTE9 and GSTS5 with allyl ITC (orange) and benzyl ITC (orange), respectively. The red dashed lines indicate the hydrogen bonds between the ITC and amino acids of the GST proteins. Significant differences between means ( ± SE) for each GST enzyme were determined by Tukey HSD tests in conjunction with one-way ANOVA in (**B**). Different lowercase letters denote significant differences (*P* < 0.05).

are also more negatively affected by aliphatic GSLs than indolic GSLs[43]. However, the piercing-sucking herbivore *Myzus persicae* (green peach aphid) is indeed negatively affected by indolic GSLs[44]. Thus, GSLs of different structures have different effects on insect herbivores, which may explain the presence of mixtures of GSLs in plants as a defense against multiple types of herbivores[45]. Although we focused specifically on how *S. littoralis* copes with ITCs, the toxicity of other products generated by the glucosinolate–myrosinase system, as well as the counteradaptation strategies employed by generalist herbivores, warrant further investigation.

Previous studies on GSL metabolism in *S. littoralis* larvae reported that the ITCs produced by hydrolysis are conjugated to GSH, with the original conjugate and several derivatives appearing in the frass[24,27,29]. Here we investigated the metabolic fate of additional ITCs, using untargeted metabolomics to look more intensively for other products. We confirmed that the major route of ITC metabolism in this insect is via GSH conjugation and subsequent steps of the mercapturic acid pathway. This is generally the principal pathway for ITC metabolism in other generalist insect herbivores,

mammals and a broad spectrum of entomopathogenic fungi[24,37,40,46]. Another amino acid-based metabolite of ITCs in *S. littoralis*, revealed by non-targeted metabolomics analyses, resulted from conjugation with the ε-amino group of lysine, forming an ITC-amine conjugate (Fig. 2). ITC-amine conjugates, such as ITC-Lys, may be more stable than GSH conjugates[47], which are known to break down under physiological conditions, especially in the presence of free thiols[46].

The formation of mercapturic acid pathway and lysine conjugates can be readily considered detoxification reactions since these processes convert very reactive hydrophobic electrophiles into less reactive, water-soluble products[21]. Previous studies with *S. littoralis* reported, however, that feeding 4MSOB ITC at 2 or 4 μmol per g in artificial diet led to a high rate of GSH conjugate formation, which depleted the supply of GSH and Cys, causing up to a one-third reduction in growth[29]. In the present study, feeding 4MSOB ITC at a lower concentration of 1 μmol per g, more typical of Brassicales leaf tissue overall, led to a significant growth reduction, but the levels of soluble protein and all measured free amino acids did not decline (Fig. 1). In fact, the

**Table 1 | Kinetic constants of *S. littoralis* GSTs with allyl ITC and benzyl- ITC**

| Enzyme | Substrate | $V_{max}$ $\mu mol \cdot mg^{-1} \cdot min^{-1}$ | $K_M$ $\mu M$ | $k_{cat}$ $s^{-1}$ | $k_{cat}/K_M$ $mM^{-1} \cdot s^{-1}$ |
|---|---|---|---|---|---|
| GSTe9 | Allyl ITC | 103.03 ± 6.38 | 61.00 ± 12.98 | 41.85 ± 2.59 | 686.05 ± 42.47 |
| | Benzyl ITC | 73.85 ± 11.28 | 109.40 ± 52.94 | 29.99 ± 4.58 | 274.17 ± 86.51 |
| GSTe14 | Allyl ITC | 89.41 ± 4.43 | 124.10 ± 15.66 | 37.43 ± 1.85 | 301.63 ± 14.94 |
| | Benzyl ITC | 47.33 ± 8.00 | 185.37 ± 89.77 | 19.82 ± 3.35 | 106.91 ± 37.29 |
| GSTe17 | Allyl ITC | 11.89 ± 0.86 | 97.16 ± 23.09 | 4.88 ± 0.35 | 50.21 ± 3.63 |
| | Benzyl ITC | 8.51 ± 2.26 | 381.65 ± 195.52 | 3.49 ± 0.93 | 9.14 ± 2.43 |
| GSTd3 | Allyl ITC | 9.43 ± 1.25 | 958.26 ± 198.40 | 3.93 ± 0.52 | 4.10 ± 0.54 |
| | Benzyl ITC | 33.55 ± 3.99 | 81.41 ± 32.92 | 13.99 ± 1.66 | 171.84 ± 50.53 |
| GSTs5 | Allyl ITC | 19.01 ± 3.26 | 198.75 ± 77.86 | 7.65 ± 1.31 | 38.50 ± 6.60 |
| | Benzyl ITC | 112.13 ± 12.75 | 252.11 ± 72.41 | 45.13 ± 5.13 | 179.02 ± 20.35 |
| GSTs6 | Allyl ITC | 39.28 ± 5.57 | 567.11 ± 131.85 | 16.10 ± 2.28 | 28.39 ± 4.02 |
| | Benzyl ITC | 52.73 ± 3.57 | 98.24 ± 21.15 | 21.61 ± 1.46 | 219.99 ± 14.91 |

Michaelis–Menten constants (± SE) were determined by nonlinear regression of activities determined using variable ITC concentrations (10 µM to 1 mM) at saturating GSH concentration (4 mM), $n = 3$ (number of replicates for each enzyme). Proteins were purified with His tags, which were not cleaved prior to assay.

**Table 2 | Binding energy and inhibition constant of ITC–GST protein interactions**

| Protein | Ligand | Binding energy $kcal \cdot mol^{-1}$ | Inhibition constant $\mu M$ |
|---|---|---|---|
| GSTE9 | Allyl ITC | −3.54 | 2550 |
| | Butyl ITC | −3.11 | 5240 |
| | Benzyl ITC | −4.57 | 446.6 |
| | 2PE ITC | −4.71 | 353.05 |
| GSTD3 | Allyl ITC | −3.01 | 6220 |
| | Butyl ITC | −3.34 | 3550 |
| | Benzyl ITC | −4.61 | 420.54 |
| | 2PE ITC | −4.43 | 568 |
| GSTS5 | Allyl ITC | −3.44 | 2990 |
| | Butyl ITC | −3.65 | 2110 |
| | Benzyl ITC | −4.76 | 322.6 |
| | 2PE ITC | −4.64 | 393.87 |
| GSTS6 | Allyl ITC | −3.93 | 1310 |
| | Butyl ITC | −4.18 | 866.38 |
| | Benzyl ITC | −4.97 | 228.68 |
| | 2PE ITC | −5.01 | 212.67 |

observed to transform benzenic GSL derivatives to glycine, alanine, serine and aspartate conjugates, while indolic GSL products were converted to glutamate conjugates[50–52]. Products of indolic GSL hydrolysis were also found to be converted by the aphid *Myzus persicae* to cysteine conjugates[44]. Meanwhile, amino acid conjugation is involved in the metabolism of other plant defense compounds. For instance, *S. littoralis* itself detoxifies the neurotoxic 3-nitropropanoic acid through conjugation with glycine, alanine, serine and threonine[53]. Together, these results identify an understudied area of insect detoxification and emphasize the importance of dietary nitrogen in influencing the ability to detoxify exogenous compounds.

While GSTs are involved in a variety of cellular functions in insects, they are best known for their roles in the detoxification of a wide range of endogenous and exogenous compounds[54]. GSTs are reported to catalyze the conjugation of GSH with electrophilic sites of diverse substances, including insecticides and plant-derived toxins like xanthotoxin, sesquiterpene lactones and ITCs[24,33,55–57]. However, the reaction pattern of the entire family of GSTs in an insect with different representatives of a class of naturally occurring plant toxins has not been previously studied. When tested with ITCs, *S. littoralis* His-tagged GSTs displayed clear preferences for reactions with ITCs with specific side chain types. For example, all GSTs with notable activity from the delta, epsilon, and sigma classes exhibited higher enzymatic activity with allyl ITC (short, unsaturated side chain) than with 4MSOB ITC (longer side chain with a methylsulfinyl group) (Fig. 3). This correlates with the significantly better performance of *S. littoralis* on allyl ITC versus 4MSOB ITC diets (Fig. 1A). Many *S. littoralis* His-tagged GSTs, especially GSTD3, GSTD2, GSTS6, GSTS5, and GSTS3, showed higher activity with benzyl ITC (shorter side chain) than 2PE ITC (longer side chain) (Fig. 3), again correlating with greater *S. littoralis* growth on benzyl ITC versus 2PE ITC diets (Fig. 1A). The same correlations are true in other organisms, although their relative activities towards ITCs may be completely opposite. For example, GSTs of the fungus *Beauveria bassiana* exhibit more activity with 4MSOB ITC than allyl ITC, and *B. bassiana* accordingly develops much better on GSL-sequestering *Brevicoryne brassicae* cabbage aphids fed on plants containing 4MSOB GSL than fed on plants containing allyl GSL[40]. To further understand the basis of these enzyme–substrate preferences, we used molecular docking simulations, which revealed that benzyl ITC and 2PE ITC formed stronger predicted interactions with GSTs, particularly GSTE9 and GSTS5, compared to allyl ITC and butyl ITC (Table 2). These structural insights support the idea that ITC side-chain structure critically influences GST binding affinity and may explain the observed differences in enzymatic activities and insect development. Together, these findings

quantities of the ITC conjugate-forming amino acids, Cys and Lys, even increased (Fig. 1C), suggesting that *S. littoralis* was able to adjust its physiology to a diet with substantial ITC content. The negative effects of feeding on ITCs thus depend not only on the actual ITC content of the diet, but also on the supply of nitrogen and sulfur-containing metabolites. For minimizing the metabolic cost of ITC processing, another feature of the mercapturic acid pathway may be important. Of the three amino acid residues constituting the GSH tripeptide, both Gly and Glu residues, but not the Cys residue, were largely recovered by reactions following initial conjugate formation, leaving the Cys conjugate as the major product (Fig. 2). These steps minimize nitrogen loss in the frass.

In a number of other insect herbivores, GSL metabolism has been reported to result in the formation of amino acid conjugates. For example, *Pieris rapae* was shown to convert benzenic GSLs to glycine conjugates via their corresponding nitriles and carboxylic acids[48,49]. More recently, two species of coleopterans, *Phaedon cochleariae* and *Galeruca tanaceti*, were

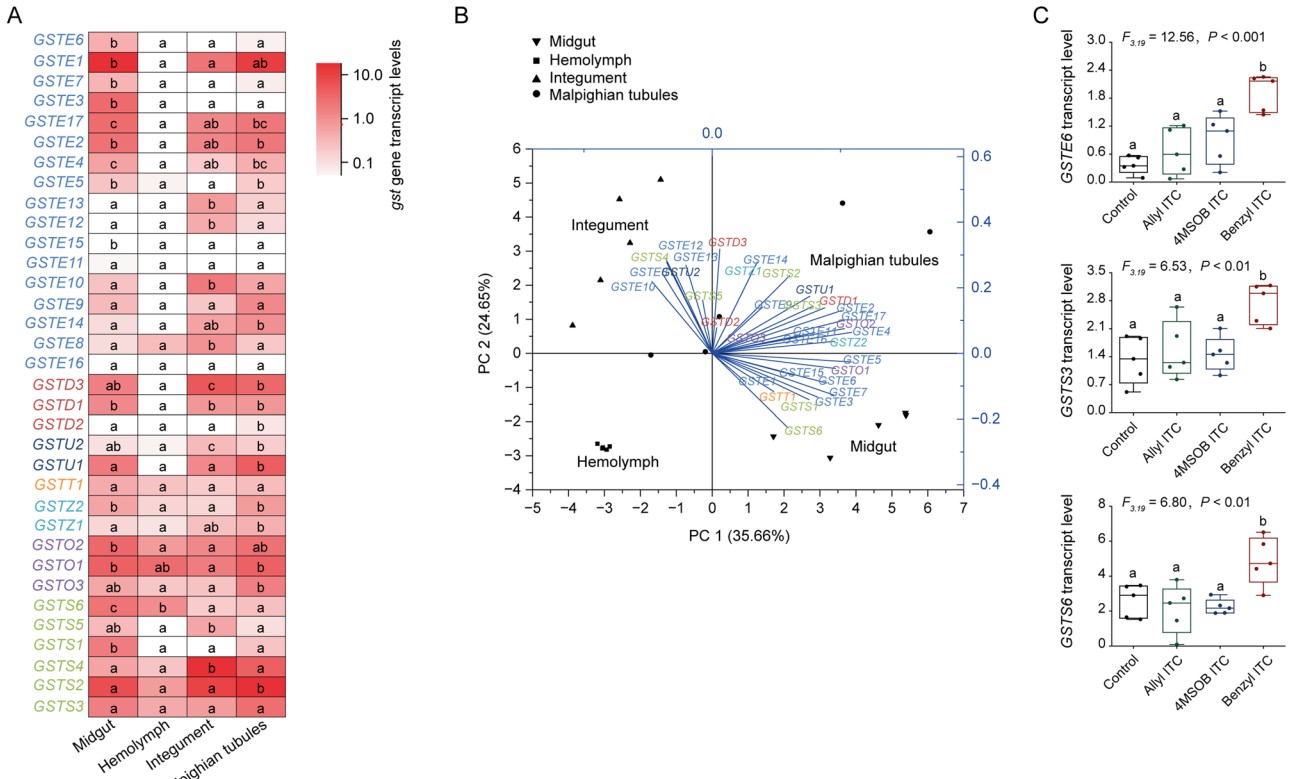

**Fig. 4 | Expression patterns of *GST* genes in *S. littoralis* larval tissues and after feeding on ITCs. A, B** Heatmap (**A**) and a PCA plot (**B**) showing the expression (relative to *RPL13*) of 34 cytosolic *GST* genes in larval midgut, hemolymph, Malpighian tubules, and integument (*n* = 5 for all points). **C** Expression levels of *GSTE6*, *GSTS3*, and *GSTS6* genes (relative to *RPL13*) induced in *S. littoralis* larval midgut by feeding on artificial diets containing ITCs (*n* = 5). The inducibility of the other *GST* genes is shown in Supplementary Fig. 8. Significant differences between means (± SE) of tissues were determined by Tukey HSD tests in conjunction with one-way ANOVA in (**A**, **C**). Different lowercase letters denote significant differences (*P* < 0.05).

underscore the significance of GST-mediated conjugation in enhancing the performance of organisms exposed to diverse toxins.

Our work also revealed striking patterns among the classes of *S. littoralis* cytosolic GSTs in terms of their reactivity with ITCs. Most of the well-studied detoxification-related GSTs reported to date belong to the delta and epsilon classes (Supplementary Fig. 7)[33]. Of the six classes whose members were studied here, three (delta, epsilon and sigma) have representatives that react with ITCs. Moreover, enzymes from the epsilon class had higher enzyme activity on unsaturated, linear, and branched aliphatic ITCs than benzenic ITCs. In contrast, GSTs from the sigma class exhibited higher enzyme activity on benzenic ITCs than aliphatic ITCs (Fig. 3B). For a single widely occurring ITC, such as allyl ITC, nearly 20 *S. littoralis* GSTs are capable of catalyzing its conjugation in vitro, although we do not know if this trend also occurs in vivo. Plant GSTs also have low specificity for their substrates[58]. In contrast, other large detoxification enzyme families often have only one or two representatives that react with a particular toxic substrate. For example, among the 123 cytochrome P450 enzymes of *Spodoptera litura*, only one could detoxify a diterpenoid antifeedant from *Nepeta stewartiana*[59]. The overlapping substrate specificity of GSTs may help insect herbivores tolerate feeding on Brassicales plants with a variety of different types of GSLs, and also on different host plants with mixtures of other electrophilic defense compounds. The broad specificity of detoxification enzymes may be especially valuable for generalist herbivores, like *S. littoralis*, that feed on a wide variety of host plants. Generalists usually have substantially larger numbers of genes encoding enzymes of the principal detoxification families (GSTs, P450s, UDP-glycosyltransferases) than specialist herbivores[60].

To test the physiological roles of specific GSTs in ITC detoxification, CRISPR/Cas9-mediated knockout of *GST* gene clusters or transgenic overexpression in *Drosophila* model systems can be useful approaches. For instance, knockout of *GSTE16* in *S. litura* significantly increased larval susceptibility to xanthotoxin, whereas transgenic *D. melanogaster* overexpressing *GSTE16* exhibited enhanced tolerance to the same compound[61]. However, GSTs constitute a large multigene family of ubiquitously expressed and polymorphic enzymes involved in the metabolism of a broad range of ITCs with diverse side-chain structures. Consequently, knockout of a single *GST* gene may not produce a measurable phenotype, as functional redundancy within the family could compensate for the loss. Despite this, gene knockout remains a powerful strategy for elucidating GST functions in the interactions between insect herbivores and plant chemical defenses. Alternatively, by leveraging the sequence similarity within *GST* subfamilies, RNA interference targeting entire *GST* classes could be employed to investigate their roles in ITC detoxification. Moreover, GSTs from the delta and epsilon classes in *D. melanogaster* have been shown to metabolize structurally distinct ITCs[36,37]. Therefore, heterologous overexpression of *S. littoralis* GSTs in a host like *Drosophila* may serve as a valuable approach to functionally validate their roles in ITC detoxification, albeit with technical and functional challenges.

The expression of genes encoding detoxification enzymes is often induced only when their substrates are present in the diet, which is usually ascribed to the energetic costs of synthesizing the enzymes when they are not needed. For instance, for the GSL sulfatase 1 of *Plutella xylostella*, which is involved in GSL detoxification, gene expression and its encoded enzyme activity are strongly induced only when this insect feeds on plants containing aliphatic GSLs[62,63]. Moreover, it has been reported that genes encoding *S. litura* GSTs from the epsilon class were also expressed at higher levels following exposure to various pesticides and plant defenses[55,64,65]. The inducibility of the GST-encoding genes of herbivorous insects by Brassicales GSLs, however, is poorly known, although it has been documented that transcripts encoding putative detoxification enzymes, including HvGSTO2

and HvGSTE11, are significantly upregulated in *Heliothis virescens* fed on plants containing GSLs[66]. Curiously, in the present study with *S. littoralis*, of the nearly 30 genes encoding GSTs that react with ITCs, *GSTS6*, *GSTS3*, and *GSTE6* were the only three induced by ITCs. All of these were induced only by benzyl ITC, not by allyl ITC or 4MSOB ITC (Supplementary Fig. 8). Appropriately, GSTS6 and GSTS3 have higher activity for benzyl ITC conjugation than allyl ITC or 4MSOB ITC (Fig. 4). The observed pattern may indicate that constitutive expression of *S. littoralis* GSTs does not incur large metabolic costs, or that it is valuable for this insect to have immediate protection against toxins without waiting for enzyme expression to be induced. Another possibility is that many *S. littoralis* GSTs function to conjugate a variety of other electrophilic toxins occurring endogenously or frequently in the diet, so that it is important to have them constitutively present.

As might be expected, the expression of detoxification-relevant genes and their corresponding enzymatic activities are frequently reported from herbivore guts, given the regular ingestion of toxins as part of the diet[62,67,68]. Within the gut, the cells of the gut epithelium are known to be sites with high levels of detoxification gene transcripts and enzymes. These cells may take up large amounts of toxins from the gut lumen or may release enzymes directly into the lumen. For example, the GSL sulfatases of *P. xylostella* are expressed in midgut cells and then secreted into the lumen to convert GSLs to desulfo-GSLs, thus avoiding GSL hydrolysis[23]. GSTs from the epsilon class of *S. litura* that help with xanthotoxin tolerance are mainly expressed in the midgut[55]. In the present study, approximately 90% of 4MSOB ITC was transformed to GSH conjugates in the midgut of *S. littoralis* larvae (Fig. 2C). The GSTs involved in ITC detoxification from the epsilon and sigma classes, such as *GSTE1*, *GSTE3*, *GSTE17*, *GSTE2*, *GSTE4*, *GSTE5*, and *GSTS6*, are all mainly expressed in the midgut (Fig. 4A).

Among other insect tissues that contact dietary toxins, the Malpighian tubules function in transporting toxins out of the hemolymph, but how much they contribute to ITC excretion is not known. However, the presence of substantially higher amounts of unmetabolized ITCs (about one-fourth of total ITCs and derivatives) in *S. littoralis* larval hemolymph, integument and frass (Fig. 2C), compared to larval gut cells (around 10%), suggests contact of Malpighian tubules with ITCs. Most of the GSTs involved in ITC detoxification in *S. littoralis*, such as *GSTE2*, *GSTE5*, *GSTE14*, and *GSTS2*, are highly expressed in the Malpighian tubules (Fig. 3B).

ITCs from the hemolymph may also contact the fat body, while the integument is vulnerable to exposure to more volatile ITCs, such as allyl ITC and benzyl ITC, that are released from the plant during feeding. Genes encoding a number of delta, epsilon and sigma GSTs, namely *GSTE13*, *GSTE12*, *GSTE10*, *GSTE8*, *GSTD3*, *GSTU2*, and *GSTS5*, were expressed at higher levels in the integument of *S. littoralis* larvae compared to other organs (Fig. 4B), with GSTD3 and GSTS5 exhibiting high enzyme activity towards the volatile benzyl ITC (Fig. 3). Thus, *S. littoralis* appears to depend on GSTs present in multiple organ systems to protect itself against toxic ITCs, although these enzymes may also function in conjugating a range of other exogenous or endogenous metabolites.

In conclusion, the GSL detoxification system of *S. littoralis* has a number of properties that seem to be appropriate for a generalist-feeding insect herbivore. After GSLs are hydrolyzed to ITCs, these are conjugated to GSH in a process similar to that found in many other generalists. However, ITC metabolism in *S. littoralis* is complex in that the enzymes conjugating the ITCs, the GSTs, each have activity with multiple ITC substrates typical of different host plants. Nevertheless, from the small diversity of ITCs tested here, *S. littoralis* GSTs would seem to have less activity with ITCs having larger, more complex side chains than those with smaller, simpler side chains. This substrate preference is reflected in the differential performance of larvae on different ITC diets. GST-mediated ITC processing in *S. littoralis* larvae is also characterized by constitutively expressed genes distributed in the gut, Malpighian tubules, and integument. These features indicate a widespread detoxification system that is always "on duty," which is a further advantage for an insect that may frequently switch its host plant.

## Methods

### Insects and plants

Eggs of *Spodoptera littoralis* Boisduval (Egyptian cotton leafworm) were generously provided by Syngenta Crop Protection (Stein, Switzerland). Larvae were hatched and reared on an artificial bean diet[29]. Larvae were maintained at 21 °C under a 12:12 h light–dark cycle. *Arabidopsis thaliana* Columbia-0 (Col-0) accession plants, which possess wild-type GSLs, were used alongside transgenic *myb28myb29* double KO mutants (lacking aliphatic GSLs)[69], *myb28myb29 × cyp79b2cyp79b3* quadruple KO mutants (lacking GSLs)[70], and *CYP79A2* KI mutants (containing benzyl GSLs and wild-type GSLs)[71]. The plants were cultivated in a climate-controlled short-day environmental chamber at 21 °C, 60% relative humidity, and a 14:10 h light–dark photoperiod. All experiments were conducted at 21 °C under a 12:12 h light-dark photoperiod. *S. littoralis* larvae at the 3rd instar stage, a stage characterized by reduced natural mortality, were used for experiments.

### GSL analysis of *A. thaliana* plants

Five-week-old *A. thaliana* wild-type, *myb28myb29* KO, *myb28myb29 × cyp79b2cyp79b3* KO, and *CYP79A2* KI plants were collected and immediately frozen in liquid nitrogen. Five replicates were conducted for each treatment. The samples were freeze-dried using an Alpha 1–4 LDplus freeze dryer (Martin Christ, Osterode am Harz, Germany) for 2 days and homogenized by shaking with 5–6 metal balls (3 mm) in each tube. GSLs were extracted from samples (approximately 10 mg leaves) using 1 mL extraction solvent (80% methanol) with 50 µM sinalbin as an internal GSL standard. After 5 min of incubation on a horizontal shaker (230 rpm) with the solvent, supernatants were collected by centrifugation (18,000×*g*). Sequentially, 800 µL of each supernatant was loaded onto a DEAE-Sephadex A-25 column. Samples on the DEAE-Sephadex were washed sequentially with 0.5 mL of 80% MeOH, followed by two washes with 1 mL of MilliQ water, and then 0.5 mL of 0.02 M MES buffer (pH 5.2). The GSLs bound on the DEAE-Sephadex were desulfated by 30 µL sulfatase (*Helix pomatia*, Sigma-Aldrich, Taufkirchen, Germany) treatment (for preparation of sulfatase solution, see Graser et al.[72]) overnight at room temperature. The next day, desulfo-GSLs were eluted with 500 µL MilliQ water for HPLC-UV measurement. Desulfo-GSLs were analyzed on an Agilent Technologies 1100 Series HPLC with a diode-array detector using a Nucleodur Sphinx RP column (250 × 4.6 mm × 5 mm, Macherey-Nagel, Düren, Germany). Desulfo-GSLs were detected at 229 nm and quantified according to Burow et al.[73]. Water and acetonitrile were employed as mobile phases A and B, respectively. The elution profile was: 0–1 min, 1.5% B; 1–6 min, 5% B; 6–8 min, 7% B; 8–18 min, 21% B; 18–23 min, 29% B; 23–24 min, 100% B; 24–28 min, 1.5% B, at a flow rate of 1.0 mL per min.

### Larval feeding assay and growth rates

In the artificial diet feeding assays for *S. littoralis*, freshly molted 3rd instar *S. littoralis* larvae were individually placed into 35 mL polystyrene Solo cups. Each larva was provided ad libitum access to a cube (~1 cm³) of artificial bean diet containing 1 µmol of ITCs per gram of diet, a naturally occurring concentration formed in Brassicales leaves after herbivore damage. In a previous study, Jeschke et al. used 2–4 µmol per g ITC based largely on data from total GSL concentration in *Arabidopsis thaliana* leaves[29]. However, most other Brassicales plants actually contain lower amounts of GSLs in their foliage than *A. thaliana*[74], and a generalist feeder such as *S. littoralis*, typically feeds preferentially on old, lower-concentration leaves rather than young, higher-concentration leaves (personal observations). Moreover, only 80–90% of the total moles of GSL present can be detected as ITC after hydrolysis[24]. Thus, we chose 1 µmol per g as a more natural concentration for bioassays of individual ITCs in this study. A negative control group was fed an artificial diet without ITCs, and two positive control groups were fed an artificial diet with 1 µmol/g insecticide (flufenoxuron and diflubenzuron). The preparation of the artificial diet for experiments was modified based on the method described in Jeschke et al.[29], which is described in the Supplementary Note 2. Each treatment comprised 15

replicates, and larvae were randomly assigned to feeding treatments. Larval weights and survival were recorded every two days throughout the experiment. During the experiment, larvae were maintained in a controlled environment at 21 °C under a 12:12 h light-dark photoperiod and were provided with a cube of fresh diet daily to minimize losses of ITCs to evaporation. Relative growth rates (RGR) were calculated for each individual caterpillar that survived until the end of the experiment. Survivorship (%) was calculated based on the percentage of surviving larvae every 2 days.

The RGR was calculated using the formula:

$$RGR = ((\ln (\text{mass of the last day}) - \ln (\text{mass of the first day}))/\text{duration of feeding in days}$$

In the plant feeding bioassays, 5-week-old *A. thaliana* Col-0 wild-type plants, *myb28myb29* KO, *myb28myb29 × cyp79b2cyp79b3* KO, and *CYP79A2* KI mutant plants were used. 3rd instar *S. littoralis* larvae were randomly assigned to plants from each treatment group. Each plant was provided with one larva and covered with a cellophane bag to prevent the caterpillar from escaping. Similar to the artificial diet feeding assay, each treatment consisted of 15 replicates. RGR was calculated based on the larval mass on the first day and the third day post-feeding for each individual caterpillar.

## Non-targeted metabolomics analyses via UHPLC-qTOF-MS

To analyze the metabolism of GSL in Brassicales by *S. littoralis*, 3rd instar larvae were fed on *A. thaliana* wild-type plants, transgenic *myb28myb29* KO mutants, *myb28myb29 × cyp79b2cyp79b3* KO mutants, and *CYP79A2* KI mutants, respectively. *S. littoralis* larvae were randomly assigned to plants from each treatment, with one larva per plant, and covered with a cellophane bag to prevent caterpillar escape. Each treatment consisted of 15 replicates. After three days of feeding, frass was collected by keeping larvae individually in 35 mL plastic cups for 6 h. Frass from 3 larvae in each treatment group was pooled as one sample, resulting in 5 replicates for each treatment. Simultaneously, the bodies of *S. littoralis* larvae (n = 5 for each treatment) were collected. Each sample was extracted with 50% methanol (in MilliQ water) in 2 mL Eppendorf tubes with 5–6 metal balls (3 mm) by vigorous shaking. Solid debris was pelleted by centrifugation, and the supernatants were transferred to new vials for analysis by UHPLC-qTOF-MS.

To analyze the effects of ITCs on *S. littoralis* metabolism, 3rd instar larvae were fed an artificial diet mix[29] containing 1 μmol per g of selected ITCs (4MSOB ITC, benzyl ITC, allyl ITC, 2PE ITC, or butyl ITC, n = 15 for each treatment). Larvae fed on an artificial diet without ITC served as a control group. Larvae from all the treatments were fed individually in 35 mL plastic cups for 3 days, with the artificial diet replaced every 24 h to ensure a fresh food supply. Frass from 3 larvae in each treatment group was pooled as one sample, and 5 larvae from each treatment were collected. Approximately 20–40 mg aliquots of each sample were extracted in 50% methanol. To analyze the metabolites in the bodies of *S. littoralis* larvae, larvae were fed an artificial diet with 1 μmol per g of 4MSOB ITC, benzyl ITC, allyl ITC, or without ITCs (n = 5 for each treatment) for 13 days. The larvae were then collected, and their bodies were extracted in 50% methanol.

The extracted compounds were analyzed by UHPLC-qTOF-MS (Dionex Ultimate 3000 series UHPLC (Thermo Scientific, Waltham, USA) and a Bruker timsToF mass spectrometer (Bruker Daltonik, Bremen, Germany). The UHPLC system was equipped with a C18 reversed-phase column (Zorbax Eclipse XDB-C18, 1.8 μm, 2.1 mm × 100 mm) maintained at 25 °C and operated at a flow rate of 0.3 mL per min with a gradient flow of 0.1% aqueous formic acid (solvent A) and acetonitrile (solvent B). The gradient profile was as follows: 5% B from 0 to 0.5 min, 5–60% B from 0.5 to 11 min, 60–100% B from 11–11.1 min, maintained at 100% B until 12 min, then re-equilibrated at 5% B from 12.1–15 min. HRMS analyses were conducted in positive ionization mode with automatic MS2 scans enabled (data-dependent acquisition). The source end plate offset was set to 500 V, capillary voltage at 4500 V, nebulizer gas at 2.8 bar, dry gas at 8 L per min, and drying temperature at 280 °C. Ion transfer was performed with specific

RF settings, and the quadrupole ion energy was maintained at 4 eV (low mass 90 m/z). The mass scan range was 50–1500 m/z at an acquisition rate of 12 Hz. Collision energies were stepped between 20 eV and 50 eV in a 50:50 timing. For recalibration of the mass spectrometer using expected cluster ion m/z values, 10 μL of a sodium formate-isopropanol solution (10 mM NaOH in 50/50 (v/v%) isopropanol-water containing 0.2% formic acid) was injected at the beginning of each chromatographic analysis. Data analysis was performed using MetaboScape 2023b software and MetaboAnalyst 6.0. Automated peak picking and alignment were conducted within a retention time range of 0.4 to 11 min, with signal intensities ≥3000 for larval frass and ≥1000 for larval bodies. The maximum deviation was set to 2 ppm. Feature groups representing single metabolites were reduced to one bucket by the MetaboScape software. Peaks not present in specific samples were assigned intensity values via k-nearest neighbors based on similar features. Data was normalized by sample weight and log-transformed base 10. Additionally, metabolomics analysis was conducted using SIRIUS (version 5.8.6) to predict molecular formulas, chemical structures, and chemical taxonomy. The Sirius import module (.mgf) file from MetaboScape software was utilized for this purpose. The .mgf file was uploaded in SIRIUS to generate predicted molecular formulas for each feature, which were then reranked using ZODIAC[75]. Predicted molecular fingerprints were generated using fragmentation trees via CSI:FingerID[76]. The chemical taxonomy of predicted metabolite structures was obtained using CANOPUS (superclass, class, and subclass)[77–79]. The results of the analysis were listed in Supplementary Data Files 1, 2, 3, and 4.

## Syntheses, purification, and NMR analyses of ITC conjugates

Confirmation of the identities of mercapturic acid pathway products, including 4MSOB ITC-GSH, 4MSOB ITC-Cys, 4MSOB ITC-Lys, and 2PE ITC-Lys, was achieved through chromatographic and MS/MS comparison with commercially available standards using UHPLC-qTOFMS. The structures of 4MSOB ITC-CysGly, 2PE ITC-GSH, 2PE ITC-CysGly, 2PE ITC-Cys, allyl ITC-GSH, and allyl ITC-Cys conjugates were compared with standards that had been previously confirmed[40]. The structure of 4MSOB ITC-Cyclic-Cys was previously confirmed elsewhere[80]. For non-commercially available mercapturic acid products such as 4MTB ITC-CysGly, 4MTB ITC-Cys, 4MTB ITC-Cyclic-Cys, benzyl ITC-CysGly, benzyl ITC-Cys, 2PE ITC-Cyclic-Cys, and butyl ITC-Cys, synthetic preparation of standards was undertaken. Specifically, 10 μL of the ITC was mixed with 25 mg of GSH, Cys-Gly, or Cys in 10 mL of water:ethanol (1:1, pH = 7) at room temperature for 48 h. The structures of benzyl ITC-Lys were also synthesized. In this process, 10 μL of the ITC was mixed with 25 mg of lysine in 10 mL of water:ethanol (1:1, pH = 10) at room temperature for 24 h.

To complete the synthesis, the ethanol in the reaction mixtures was evaporated under N₂ overnight. The ethanol-free mixture underwent purification through a solid-phase extraction (SPE) C18 column (5 g C18, 45 mL volume) previously conditioned using pure methanol followed by MilliQ water. Subsequently, the SPE C18 column cartridge was subjected to a stepwise gradient of aqueous methanol solutions (ranging from 0%, 5%, 10%, 15%, 20%, 30%, 40%, 50% to 100% methanol in water; v: v). The concentrated methanol solution containing the targeted fragment underwent further purification via repeated injection into an HPLC-UV system (Agilent Technologies 1100 Series HPLC) equipped with a diode-array detector, using a Nucleodur Sphinx RP column (250 × 4.6 mm × 5 μm). Detection wavelengths were set at 220 nm for ITC-Cys and ITC-Cyclic-Cys conjugates, 280 nm for ITC-CysGly conjugates, and 240 nm for ITC-Lys conjugate. Mobile phases A and B comprised aqueous formic acid (0.1%) and acetonitrile, respectively. The elution profile was: 0–8 min, 5–45% B; 8–8.10 min, 45–100% B; 8.10–10 min, 100% B; 10–11.1 min, 100–5% B and 11.1–15 min, 5% B at a flow rate of 1.0 mL per min. The system was equipped with a fraction collector (Advantec SF-2110), and the collected conjugates were concentrated using a rotary evaporator. The structures of all purified ITC conjugates were confirmed by NMR spectroscopy (Supplementary Note 1 and Supplementary Fig. 4). The presence of these ITC

conjugates in the synthesized samples was verified using an HPLC (Agilent HP1100 series, Agilent Technologies) ion trap mass spectrometer (ESQUIRE-6000 system, Bruker) using a Nucleodur Sphinx RP column ($250 \times 4.6$ mm $\times 5$ µm). Aqueous formic acid (0.2%, solvent A) and acetonitrile were employed as mobile phases A and B, respectively. The elution profile was: 0–20 min, 10–50% B; 20–20.10 min, 50–100% B; 20.10–22 min, 100% B; 22–22.1 min, 100–10% B; and 22.1–26 min, 10% B at a flow rate of 1.0 mL/min. MS analyses were performed separately with positive and negative ionization and automatic MS2 scans ("autoMS") enabled. The following parameters were used: capillary exit voltage, $+117/-117$ eV; capillary voltage, $+3000/-3000$ V; nebulizer pressure, 35 psi; drying gas, 11 l min-1; gas temperature, 330 °C. The mass scan range was 60–1000 $m/z$.

NMR measurements were performed on a 500 MHz Bruker Avance III HD spectrometer equipped with a TCI cryoprobe using standard pulse sequences as implemented in Bruker Topspin ver. 3.6.1 (Bruker Biospin GmbH, Rheinstetten, Germany). Samples were measured in MeOH-$d_3$ or $D_2O$, depending on their solubility. The samples were carefully tuned and matched prior to the measurements. Chemical shifts were referenced to the residual solvent signal of MeOH-$d_3$ ($\delta_H$ 3.31/$\delta_C$ 49.0). No chemical shift correction was applied for measurements in $D_2O$. NMR spectra of 4MTB ITC-Cyclic-Cys and 2PE ITC-Cyclic-Cys were recorded at 238 K to observe all carbon signals. All other spectra were recorded at 298 K.

### Targeted HPLC–MS/MS analyses of 4MSOB ITC metabolites
To quantify the 4MSOB ITC metabolites of *S. littoralis*, 3rd instar *S. littoralis* larvae ($n = 15$) were fed an artificial diet mix containing 1 µmol per g of 4MSOB ITC for 3 days. Tissues from three larvae were pooled to produce one sample in 2 mL Eppendorf tubes, with 5 replicates conducted for each treatment. After 3 days of feeding, artificial diet, midgut, hemolymph, integument, and frass were collected. Hemolymph was obtained using a 10 µL pipette through a small wound scratched by a 5 mm needle. Midguts were dissected in TE buffer (Tris-EDTA buffer, pH 8.0) under a dissecting microscope, and the midgut content was carefully removed. Dissected midgut epithelium was then carefully washed in TE buffer to remove any adhering food material and hemolymph. Similarly, dissected integument was washed in TE buffer before collection. Larval frass and artificial diet were collected with 5 replicates. All collected larval tissues, frass, and artificial diet were weighed, immediately frozen in liquid nitrogen, and stored at 80 °C until further use.

To quantify the conversion of 4MSOB ITC into its conjugates by *S. littoralis*, larvae ($n = 20$) were individually placed in 35 mL polystyrene Solo cups and starved for 20 h. Each larva was then provided ad libitum access to a cube ($\sim 1$ cm³) of artificial bean diet supplemented with 1 µmol of 4MSOB ITC per gram of diet. Five diet cubes were placed in empty cups as controls to determine the initial weight of the diet provided. After 24 h of feeding, the remaining diet was collected, along with the diet cubes from the control cups. Larvae were then maintained in the same cups for an additional 3 h to allow for excretion. Subsequently, frass and larvae were collected separately. All samples were freeze-dried and weighed. The amount of diet consumed was calculated by subtracting the weight of the remaining diet from the average initial weight of the control cubes. Each sample was extracted in 1 mL of 60% methanol and analyzed by LC–MS/MS.

The analyses of 4MSOB ITC and its conjugates were conducted using an Agilent HP1260 Series instrument coupled to an API5000 tandem mass spectrometer (AB Sciex, Darmstadt, Germany) by loading extracted samples onto an Agilent Zorbax Eclipse XDB-C18 column ($50 \times 4.6$ mm $\times 1.8$ µm, Agilent Technologies, Waldbronn, Germany) with mobile phase A (0.05% formic acid in milliQ water) and mobile phase B (acetonitrile). The elution profile was: 0–0.5 min, 3% B; 0.5–2.5 min, 15% B; 2.5–2.6 min, 85% B; 2.6–3.6 min, 100% B, 3.6–6 min, 3% B at a flow rate of 1.1 mL per min. Quantification of each compound was achieved by multiple reaction monitoring (MRM) of specific precursor to product ion conversions for each compound. Parameters are described in Supplementary Table 3. Analyst 1.5 software was used for data acquisition and processing. Quantification of individual compounds was achieved by external calibration

curves. The origin of the external standards is listed in the Supplementary Table 4.

### Soluble protein and amino acid content
*S. littoralis* larvae that were fed an artificial diet containing 1 µmol per g of 4MSOB ITC, benzyl ITC, and allyl ITC for 13 days were subjected to analysis. After freeze-drying and weighing, *S. littoralis* larvae were extracted in Tris buffer (50 mM, pH = 7.5) for soluble protein, amino acid, GSH and GSSG measurement ($n = 5$ for each treatment). The total amount of soluble protein was measured using a Bradford assay. A volume of 5 µL of clear supernatants from each sample was used to measure protein concentrations using Quick Start Bradford 1× Dye Reagent. Bovine serum albumin served as a protein external standard. GSH and GSSG were analyzed immediately after extraction. Cys and Gly in the aqueous extracts were measured as their FMOC derivatives using an LC–MS/MS system. The FMOC-derivatization process involved mixing 10 µL of the aqueous extract with 90 µL $^{13}$C- and $^{15}$N-labeled amino acid standard solution (20 µg per mL) and 100 µL borate buffer (0.8 M, pH 10). Subsequently, 200 µL FMOC-reagent (30 mM FMOC-Cl in acetonitrile) was added, and the reaction was gently mixed and incubated for 5 min. Excess FMOC-Cl was removed by extraction with hexane (800 µL), and after phase separation, 100 µL of the bottom aqueous phase was collected and transferred to a glass vial. The analysis of GSH and GSSG, as well as FMOC-derived amino acids, was performed on an Agilent HP1260 Series instrument coupled to an API5000 tandem mass spectrometer (AB Sciex). GSH and GSSG were analyzed by loading samples onto a Nucleodur Sphinx RP column ($250 \times 4.6$ mm $\times 5$ µm) with mobile phase A (0.2% formic acid in milliQ water) and mobile phase B (acetonitrile). The elution profile was: 0–9 min, 2% B; 9–9.1 min, 35% B; 9.1–11.1 min, 100% B; 11.1–15 min, 2% B at a flow rate of 1.0 mL per min. FMOC–derivatization products were analyzed by loading samples onto an Agilent Zorbax Eclipse XDB-C18 column ($50 \times 4.6$ mm $\times 1.8$ µm) with mobile phase A (0.05% formic acid in milliQ water) and mobile phase B (acetonitrile). The elution profile was: 0–4.5 min, 10% B; 4.5–6 min, 90% B; 6–6.51 min, 100% B; 6.51–9 min, 10% B at a flow rate of 1.1 mL per min. Remaining amino acids were measured by mixing 10 µL of the aqueous extract with 90 µL $^{13}$C- and $^{15}$N-labeled amino acid standard solution (10 µg per mL) and analyzed using an Agilent 1260 HPLC coupled to an API6500 tandem mass spectrometer. HPLC was equipped with an Agilent Zorbax Eclipse XDB-C18 column ($50 \times 4.6$ mm $\times 1.8$ µm) with mobile phase A (0.05% formic acid in MilliQ water) and mobile phase B (acetonitrile). The elution profile was: 0–2.7 min, 3% B; 2.7–3.1 min, 100% B; 3.1–6 min, 3% B, at a flow rate of 1.1 mL per min. Quantification of each compound was achieved by multiple reaction monitoring (MRM) of specific precursor to product ion conversions for each compound. Parameters are described in Supplementary Table 3. Analyst 1.5 software was used for data acquisition and processing. Quantification of individual compounds was achieved by external or internal calibration curves, and the origin of the external and internal standards is listed in Supplementary Table 4.

### RNA isolation, cDNA synthesis and quantitative RT-qPCR
To detect the expression levels of *GST* genes in *S. littoralis* larval midgut, hemolymph, integument, and Malpighian tubules, 3rd instar *S. littoralis* larvae fed on artificial diet were collected and pooled into TRIzol reagent, and then stored at 4 °C before use. The inducibility of *S. littoralis* *GST* genes was detected in the midgut of 3rd instar *S. littoralis* larvae fed on an artificial diet supplemented with 4MSOB ITC, allyl ITC, or benzyl ITC for 2 days. *S. littoralis* larvae fed on an artificial diet without ITC served as the control group. As described above, midgut samples were collected and pooled into TRIzol reagent. Total RNA was isolated from the stored tissues according to the manufacturer's protocol. Genomic DNA contamination was eliminated using RNase-free DNase. The quantity and quality of RNA samples were examined using a NanoDrop 2000c spectrophotometer. cDNA was synthesized from the extracted RNA using SuperScript III Reverse Transcriptase kits. RT-qPCR was performed to measure gene transcripts in these cDNA samples using Brilliant III Ultra-Fast SYBR Green QPCR Master

Mix. The transcript levels of *S. littoralis* larval *GST* genes were measured by normalizing to the expression of the *S. littoralis rpl13* gene (housekeeping gene). All gene information and accession numbers are listed in Supplementary Table 5. Primer pairs were designed using Geneious Prime software.

### *S. littoralis GST* gene cloning and recombinant GST protein expression

To determine the specific activity of *S. littoralis* GSTs with ITCs, GST proteins were recombinantly expressed in *Escherichia coli* cells. The complete coding sequence of *S. littoralis GST* genes was obtained from Durand et al.[32]. The full-length target *GST* mRNA sequences were cloned from the synthesized cDNA pool using primer pairs FLF and FLR (refer to Supplementary Data 5). The full-length sequences of *GST* genes, determined by Sanger sequencing, were submitted to NCBI (accession numbers from PQ722158 to PQ722184 listed in Supplementary Table 5). A phylogenetic analysis of *S. littoralis* GST proteins was constructed from the amino acid sequences of 34 GSTs. The analysis was conducted using the neighbor-joining method in the Clustal Omega 1.2.2 program via Geneious Prime software. The scale bar represents 0.3 expected amino acid substitutions per site.

To produce recombinant *S. littoralis* GST proteins, the relevant restriction enzyme cutting sites were added to the ends of the full-length *GSTs* using the primer pairs VF and VR, and the fragments were digested by restriction enzymes. The pET28a vector used to express the target protein was also digested with the relevant restriction enzyme. The restriction-digested *GST* fragments were inserted into the restriction-cut pET28a cloning site using T4 DNA ligase. The inserted vectors were then transformed into *E. coli* BL21 (DE3) star cells using chemical transfection. Selected colonies were incubated in 5 mL LB medium with 80 µg per mL kanamycin overnight at 37 °C. Then, 1 mL of the culture was added to 100 mL LB medium with 80 µg per mL kanamycin at 37 °C for 3 h until reaching an $OD_{600}$ value of 0.5-0.8. Subsequently, IPTG (1 mM) was added to induce protein expression overnight at 18 °C with agitation at 200 rpm. Simultaneously, *E. coli* BL21 (DE3) star cells containing the pET28a empty vector were used as a negative control. The histidine-tagged recombinant GST was affinity-purified using Ni-NTA agarose resin as follows. *E. coli* cells collected in 50 mL Falcon tubes were centrifuged ($13,000 \times g$ at 4 °C for 30 min) to remove the supernatant. The pellet was resuspended in 1.5 mL lysis buffer (50 mM Tris, 20 mM imidazole, 500 mM NaCl, 10% glycerol, and 0.5% Tween 20; pH 7.5) containing protease-inhibitor mix HP (1:100 v/v) and benzonase (2 µL per 10 mL lysis buffer), and then incubated on ice for 30 min before sonication using an ultrasonic homogenizer (Sonoplus HD 2070, Bandelin, Berlin, Germany). The supernatant of lysed cells was collected by centrifugation ($13,000 \times g$ at 4 °C for 30 min) and transferred to cleaned Ni-NTA agrose resin in 2 mL Eppendorf tubes. The binding of the histidine-tagged protein to Ni-NTA agarose resin was carried out thoroughly by mixing under circular rotation at 4 °C for 1 h. The collected Ni-NTA agarose resin was washed twice using wash buffer (50 mM Tris, 20 mM imidazole, 500 mM NaCl, and 10% glycerol; pH 7.5). Afterwards, the GST protein was eluted from Ni-NTA agarose resin using elution buffer (50 mM Tris, 250 mM imidazole, 500 mM NaCl, and 10% glycerol; pH 7.5) and the buffer was changed to potassium phosphate buffer (50 mM, and 10% glycerol; pH 6.5) using Amicon ultra-10 K centrifugal filter units. The purity of the eluted recombinant His-tagged proteins was analyzed by SDS-PAGE (Supplementary Fig. 9).

### Enzyme assay and kinetic assay

Screening assays and kinetic assays were conducted in 500 µL volumes in a micro-cuvette (104.002B-QS 10 mm), and absorbance was measured using a spectrophotometer at 274 nm and 25 °C, and compared to the absorbance of no-enzyme controls ($n = 3$ for each treatment). The linear reaction rates were corrected for the non-enzymatic reaction rate and converted to molar turnover rates using published

extinction coefficients for ITC-GSH conjugates (allyl ITC-GSH $\mathcal{E}_{274} = 7.45$ mM$^{-1}$ cm$^{-1}$; butyl ITC-GSH $\mathcal{E}_{274} = 7.75$ mM$^{-1}$ cm$^{-1}$; iso-butyl ITC-GSH $\mathcal{E}_{274} = 8.11$ mM$^{-1}$ cm$^{-1}$; *sec*-butyl ITC-GSH $\mathcal{E}_{274} = 7.79$ mM$^{-1}$ cm$^{-1}$; 3MSOP ITC-GSH $\mathcal{E}_{274} = 8.00$ mM$^{-1}$ cm$^{-1}$; 4MSOB ITC-GSH $\mathcal{E}_{274} = 8.00$ mM$^{-1}$ cm$^{-1}$; benzyl ITC-GSH $\mathcal{E}_{274} = 9.25$ mM$^{-1}$ cm$^{-1}$; 2PE ITC-GSH $\mathcal{E}_{274} = 8.89$ mM$^{-1}$ cm$^{-1}$)[35,81]. GSH and ITCs were freshly prepared before the assay and dissolved in potassium phosphate buffer (100 mM, pH 6.5) and 60% acetonitrile, respectively. For the screening assays, a final concentration of 1 mM GSH, 0.4 mM ITC (in 2.4% acetonitrile), and 1–5 µg purified His-tagged GST protein in a 500 µL potassium phosphate buffer (100 mM, pH 6.5) was used. For the kinetic assays, a final saturating GSH concentration of 4 mM, variable ITC concentrations ranging from 10 µM to 1000 µM in 2.4% acetonitrile, and 1–5 µg His-tagged GST proteins in a 500 µL potassium phosphate buffer (100 mM, pH 6.5) was used. Michaelis-Menten kinetic constants (±SE) were determined for ITCs by nonlinear regression using SigmaPlot 14.0.

### Molecular docking

Molecular docking studies were performed by applying AutoDock Tools (V 1.5.6, Molecular Graphics Laboratory, Scripps Research Institute). Crystal structures of GSTs were acquired from the AlphaFold protein structure database (https://alphafold.ebi.ac.uk/). Detailed information about the protein structures used from AlphaFold is listed in Supplementary Table 5. Prior to the docking simulation, the missing atoms were replaced, hydrogen atoms were added to each structure, and then the charges were added. From the residues known to be active in the interaction of GSH with previously-determined structures of *Musca domestica* epsilon-GST (PDB https://doi.org/10.2210/pdb3vwx/pdb), *Nilaparvata lugens* delta-GST (PDB: https://doi.org/10.2210/pdb3wyw/pdb), and sigma-GST (PDB: https://doi.org/10.2210/pdb5h5l/pdb), the putative binding sites of *S. littoralis* GSTs were hypothesized[82–84]. The grid box centered on the active sites was set up with a map of $60 \times 60 \times 60$, with a spacing value of 0.5. Later, the docking calculations were carried out with 50 independent Genetic Algorithm runs and a population size of 300. Finally, all docking results and 3D geometries were observed using the AutoDock Tools and PyMOL Molecular Graphics system (V 3.1.4.1, Schrödinger, LLC).

### Statistics and reproducibility

For all statistical analyses, Origin 2023b was used, where the data distribution was examined with the Kolmogorov–Smirnov test for two-sample comparison and the group variance was examined with Levene's test for multiple comparisons. Kaplan-Meier survival estimates were computed in Fig. 1 and Supplementary Fig. 1. Multiple comparisons were performed with one-way or two-way ANOVA in Figs. 1–4, as well as Supplementary Figs. 1, 2, 6, 7, and 8, with statistical significance set at the 0.05 level. A two-tailed Student's t-test was used in Fig. 2 and Supplementary Fig. 3. Comparisons were performed with pair-wise PERMANOVA in Fig. 2 and Supplementary Fig. 6. Significant differences were determined by a paired-sample Wilcoxon signed-rank test in Supplementary Fig. 7. In graphs, data were analyzed as mean ± SE. All data presented are representative of at least three independent experiments, as indicated in the figure legends, along with the number of biological replicates. Different lowercase letters in the graphs represent $P < 0.05$, and asterisks represent: n.s, $P \geq 0.05$; *, $P < 0.05$; **, $P < 0.01$; ***, $P < 0.001$.

### Reporting summary

Further information on research design is available in the Nature Portfolio Reporting Summary linked to this article.

### Data availability

All raw data supporting the findings of this study are available through Edmond, https://doi.org/10.17617/3.WAOR7H[85], the Open Access Data Repository of the Max Planck Society, as well as in the Supplementary Data 8

and Supplementary Information file. The accession number of GST genes (PQ722158-PQ722184) reported in this paper are available in the NCBI database. All other data are available from the corresponding authors on reasonable request.

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

## Acknowledgements
We thank Prof. Ute Wittstock (Technische Universität Braunschweig, Braunschweig, Germany) for generously providing *Arabidopsis thaliana CYP79A2* knock-in mutant seeds. We thank the members of the Department of Biochemistry at the Max Planck Institute for Chemical Ecology for their helpful advice and technical assistance, especially Bettina Raguschke for DNA sequencing. We are grateful to Andrea Lehr (Research Group Plant Defense Physiology) for rearing *Spodoptera littoralis*, Xingcong Jiang (Department of Evolutionary Neuroethology) for useful comments on the manuscript and the greenhouse team for providing fresh plants. This research was supported by funding from the Max Planck Society.

## Author contributions
R.S., J.G., and D.G.V. together conceived the project, designed the experiments and interpreted the results. R.S., S.R., and B. R. conducted *Spodoptera littoralis* feeding assays and chemical extractions, Y.N. conducted NMR analyses for synthesized compounds, R.S., M.R., and D.T.M. contributed to chemical analyses, R.S., S.R., and K.L. designed protein recombination, R.S. performed molecular docking simulations, R.S., J.G., and D.G.V. wrote the paper, and all authors read, edited, and approved the paper for publication. R.S., J.G., and D.G.V. co-supervised the project.

## Funding

## Competing interests
The authors declare no competing interests.
