## [Peer Review file · Communications Biology]

Multiple glutathione-S-transferases detoxify diverse glucosinolate-based defenses of Brassicales plants in a generalist lepidopteran herbivore (*Spodoptera littoralis*)

Corresponding Author: Dr Ruo Sun

Version 0:

Reviewer comments:

Reviewer #1

(Remarks to the Author)

The authors present a comprehensive and well-executed study on the detoxification mechanisms of glucosinolate-derived isothiocyanates (ITCs) in *Spodoptera littoralis*, with a particular focus on the role of glutathione-S-transferases (GSTs). By integrating toxicological experiments, metabolomics, biochemical assays, and gene expression analysis, the study provides a holistic view of ITC detoxification in *S. littoralis*. The production of total 34 recombinant GSTs and the subsequent enzymatic assays effectively demonstrate the capacity of multiple GSTs to catalyze ITC-GSH conjugation, highlighting both the redundancy and broad substrate specificity of these enzymes. Overall, this work is resource-intensive and could be accepted after minor revisions.

When larvae were fed artificial diets containing ITCs, could the authors estimate the overall percentage of ITCs that were converted? This data is important for understanding the efficiency of the detoxification process.

The study lacks genetic validation (e.g., CRISPR knockout of GST gene clusters or overexpression in *Drosophila* model systems) to confirm the physiological role of specific GSTs in ITC detoxification. The authors should acknowledge this limitation and discuss how future genetic studies could further validate the functional importance of GSTs in vivo.

The comprehensive GST enzymatic activity data against different ITCs is a valuable resource. Would it be possible to employ AlphaFold3 or similar AI-based models to gain structural insights that correlate with these data? A brief discussion of this possibility would enhance the study.

Minor Comments:

Figure 1A-B: There is no positive control in the bioassays (e.g., a known growth inhibitor such as azadirachtin) to provide a reference for the relative toxicity of these ITCs. Including such a control would strengthen the interpretation of the results.

Figure 1D: The manuscript should clarify whether the *myb28myb29;cyp79b2cyp79b3* mutant is a transheterozygote or a double null allele.

Reviewer #2

(Remarks to the Author)

Versatile GST based detoxification Sun et al.

This is an important, interesting and well carried out investigation of the complex, general biochemical ways a generalist insect detoxifies a typical complex array of isothiocyanates from crops (such as mustards, canola and cabbages) and wild plants. It seems to be carried out with an open mind for complexity, and at relevant concentrations for the typical plant levels, as opposed to some previous studies carried out at somewhat higher concentrations. I have few major objections to design, description, discussion and interpretation of the experiments, except a method concern, a discussion concern, and a

concern that the final conclusion is a bit too strong.

Conclusion-concern: From six relatively equal sized ITCs, with no replication of the types of ITCs (only one unsaturated, only one SO-containing, modest size difference), I do not agree that one can conclude (line 397-399) "S littoralis GSTs generally have less activity with ITCs having larger, more complex side chains as those with smaller, less complex side chains". This statement is bound to be copied and cited without proper caution. I suggest to reword, e.g. "from the small diversity of ITCs tested here, S. littoralis GSTs would seem to have less activity with ITCs having". Also same section line 393-394, I would write: "S. littoralis has a number of properties that seems to be appropriate for a generalist..." (seems to be instead of are). I tend to agree, but who are we to judge?

The method concern is this: You calculate GST activity from absorbance change at 274 nm, using extinction coefficients of the substrates. But what is the extinction coefficient of the GST-ITC product at this wavelength. If the product has significant absorption, the initial rates would be under-estimated, it seems. Please could you comment on this in the revised version and adjust calculations if needed?

The major discussion concern is this: Line 251-252: "isothiocyanates have been found to be the major toxic hydrolysis product derived from aliphatic GSLs" with a general ITC review as reference. But there are five major kinds of hydrolysis products from aliphatic GSLs: nitriles, ITCs, OZTs, epithionitriles and thiocyanates, as Jena-researchers and others have so beautifully described over the years. Do you trust this very general review for stating that ITCs are the major toxic products? Here, a broader view and some context would be appropriate, to make sure that the outsider don't leave your paper with the impression that ITCs are the essence of GLs. In this paper, you have studied how a generalist copes with just one of the many weapons crucifers use for defending themselves. There is much to do still before we understand how generalists cope with the GL-MYR system.

The abstract and title I agree with, neither exaggerate the findings, and the major concerns mentioned above do not relate to these parts. But could you consider whether a revised title could include that a very large number of GSTs seemed to be involved, as I find this to be a significant finding.

Minor:

In contrast, I have numerous minor points. In all cases, I would trust the authors to be the final authority as to follow my advice or not, but I ask the authors to consider the following points.

Hyphens or not. Since isothiocyanates are spelled in two words in chemistry (allyl isothiocyanate), I object to using hyphens in the abbreviations. It should be allyl ITC. Not allyl-ITC. This requires modification of all text and all figures.

Gene abbreviations. The authors seem to consistently spell genes with standard font. But don't we usually abbreviate genes with italics and capitals (CYP79A2, GSTE2), and only use normal font for genotypes lacking that gene? If you agree, the knock in plants (e.g. line 179) and all reference to GST genes (e.g. line 238, line 387-388) should be capitalized. If you agree, this is a problem throughout the text in numerous places.

Km or KM: I believe capital M is correct. The author who made Table 1 seems to agree. So I think you should capitalize M line 216.

Testing of His-tagged proteins: Please make it crystal clear in abstract and everywhere in text, that the GSTs you tested for enzyme activity were not the native GSTs but His-tagged derivatives. His tags can be removed, but you like most other authors assumed it not needed. Future scientists may disagree, we can at least be very specific in pointing out whether derivatives or native enzymes are tested. Please do this consistently and in abstract, not only in methods.

Precise use of references. In the introduction, I several times (but not always) felt that references were used in a way not entirely loyal to the primary conclusions of those references. I also wondered whether some "Jenacentrism" had unconsciously been involved in selecting references. Please consider the following cases and reword or find other references as appropriate, if you agree with my concerns:

Line 30: I am aware that strict, careful counting has limited the number of concluded GLs to around that number, but do you agree that solid evidence for the existence of many more has been provided? Even if those additional are not 100% confirmed to structure, I think we can agree that there are many more, yet to be elucidated, than the current count. Can you in some way include this (e.g. "with c. 100 currently elucidated structures" or similar)?

Line 34: ref 8: is this really a new, state of the art reference for evolutionary diversification of GLs? (or is it the one that first comes to mind a Jena-based scientists?)

Line 35, ref 9: Does the cited ref really back up this statement? To me, ref 9 seems to deal with the opposite problem, how plants with different GL profile influence the associated fauna...

Line 45: ref 10, do you find this to be a primary, authoritative reference on this specific topic? Ref 12: this seems to deal with using ITCs to cure cancer, quite something else than preventing cancer... Ref 13: ok.

Line 47-51: You write "Most research on the biological effect of ITCs has focused on those present in common crucifers, but has not looked at a variety of chemical structures", and cite a review and a Jena paper that indeed only looked at a few. Although it is true that most papers test only one or a few ITCs, there are many excellent papers systematically testing a large number of GL products (GL+myrosinase) at conditions favoring ITCs. You actually cite one of them later (ref 46). I suggest to rephrase this section and provide the reader with the information that we have studies much larger than the present systematically testing structure-activity relationships (most based in the Bologna GL tradition).

This concern also goes for the next sentence. You cite three papers (19-21, two from Jena) for our lack of ability yet to make generalizations. But what if you consider the Bologna papers in addition?

Line 51. Natural ITCs are not "typically lipophilic". But sure, you decided to test mainly lipophilic ones. Please reconsider wording.

Line 52. The inside of the cell is not an environment (environment = surroundings of the organism, right?). Intracellular space?

Line 54-55. This statement seems to prepare the reader with a possible explanation of higher activity of ITCs with a sulphanyl group. But you never catch up on this angle in the discussion. And do you really think that a sulphanyl could influence the ITC reactivity over a distance of four methylene groups? I'd suggest to delete this sentence and reference.

Line 55-56: This is wrong in my opinion, structure activity relationships have been investigated in detail in several systems (see above concerning line 47-51).

Line 684: If $P = 0.05$, your key will both result in n.s. and *, please revise.

Figures

All. The final line in all legends, explaining the meaning of lowercase letters, need re-writing. Of course I can guess what you mean, but you really don't explain how to interpret the lowercase letters (I guess results NOT having the same lowercase letter but only different lowercase letters, are significantly different).

Fig. 1A 4MSOB, the text is formally wrong, it is a methylsulfinylalkyl side chain, isn't it?

Fig. 1B. How can I understand which curve represents which ITC, when there are three curves with three colors but seven colors in the key? I guess the experiments fell into three groups, perhaps you could state that blue indicates this, this and this, red indicates that, and that, green indicates....

Fig. 1D I was missing a color indicating "other" I guess there were a bit of other GLs in the leaves?

Fig. 2A,D, E. It took me a while to understand how the list of metabolites in red represents specific dots (the text at dots is VERY small). If you can in some way make this more clear, it would be fine. But if not, a reader like me will eventually get it.

Reviewer #3

(Remarks to the Author)

The ability to detoxify compounds in the environment (especially those in high concentrations in the diet) is key to persistence by ecological generalists, such as crop pests with broad host ranges. This is a complex process that may involve a diversity of related compounds and a diversity of related enzymes. Sun et al. investigate this process in a generalist lepidopteran that can feed on plants producing a variety of related glucosinolate compounds (precursors to toxic isothiocyanates), and which encodes dozens of enzymes potentially capable of catalyzing their detoxification (glutathione S-transferases). Their work gives novel and well-supported insight into this process, and how it differs from simple textbook cases of highly specific detoxification of individual compounds by individual enzymes with high specificity. It's important to appreciate the potential for less specific processes as well, and this paper makes an excellent case.

The authors find a number of intriguing patterns, such as a rough correlation between larval performance on various isothiocyanate-containing diets and overall GST activity against those isothiocyanates, supporting the overall conclusion that "the propensity of multiple GSTs to react with an individual isothiocyanate and the wide expression of GST-encoding genes across larval organs likely support this generalist herbivore's ability to thrive on glucosinolate-defended Brassicales plants." This is an appropriate conclusion backed by a tremendous amount of work: characterizing the impact of particular toxins on larval performance, the accumulation of these toxins and the exact compounds produced by their metabolism in vivo, and the (inducible) expression and activity and substrate specificity of the entire repertoire of GST enzymes potentially capable of catalyzing this metabolism.

Particularly impressive was the rigor of the methods employed in this paper. For example: going above and beyond to include standards in their chemical profiling, even when this meant synthesizing standards that weren't available commercially; or utilizing both artificial diets and Arabidopsis glucosinolate mutants to investigate the effect of particular classes of glucosinolates.

I enjoyed the paper and think it will be an excellent contribution to the field. I just have a few revisions to suggest, relatively minor overall. Importantly, care must be taken to ensure that all data and relevant code or "roadmaps" to the analysis are shared to make the figures reproducible (point #1).

1. Data availability: I couldn't find the original data that went into creating many of the main figures, and it was not straightforward to search through the supplementary tables for the data I hoped to find (which I did not always successfully find). For example, for Figure 1, I found supplementary tables of fold changes for each LC-MS/MS feature but not a table of all detected intensities for all features across replicated measurements. I did see a note that all data will be made available upon publication. As part of this, I would like to in the supplement see a list of the input data files used for each figure panel, as well as a general roadmap to how that data was plotted and statistically analysed to create that figure panel, to make the figures reproducible. This would be not only a helpful resource for future readers, but an exercise to ensure that all necessary data is included. Preferably, I would like to see this input list / roadmap in any revised submission, as well as offering access for reviewers to actually review additional raw data that hasn't been shared yet and will be housed on external repositories so that they can assess completeness. I am open to other approaches but overall I do strongly feel it's important to make data more complete and easier to navigate, and to make the figures more easily reproducible.

2. Figures (general): the text is quite small, and I had to zoom in a lot on my computer. But the captions were taking a lot of space, and this may not be true in the final manuscript (where each figure could take up a full page, perhaps?). Please think carefully about if the text needs to be larger.

3. Fig. 1A: What is the duration of the experiment for RGR (how many days passed)? Was it the same for all ITCs? It would be helpful to note this clearly in the caption.

4. Fig. 1B: I cannot see most of the colored lines that should be there, based on the legend showing 7 lines.

5. Fig. 1A,D,E: I can't tell which points some of these labels correspond to. Please use lines to indicate the relationship between words and points. Or perhaps plotting a different symbol for identified and unidentified/unlabeled points would be sufficient.

6. L158-159: "Most of the 4MSOB-ITC and its derivatives were excreted in the frass with less than 1% remaining in the hemolymph (Figure 2C)." Unless I'm missing something, this is not the right experiment/analysis to make such a conclusion, or otherwise very specific logic must be provided to account for how the ITC and its derivatives were proportionately tracked through the organism. The figure shows concentrations of each compound in each tissue/substance. But there is much more hemolymph than frass, and this doesn't seem to be accounted for (e.g., what is the hemolymph mass and the total mass of frass produced?). So even with a very high concentration in the frass, a substantial amount (more than 1% of what's ingested) could remain in the hemolymph or other tissues. Please be more precise with logic or adjust the statement to simply refer to concentrations.

7. L205: It's a bit hard to see the support for this. Perhaps these comparisons could be plotted in the supplement and evaluated with a paired-sample Wilcoxon signed-rank test (with each GST being a "sample")?

8. L230: Maybe I'm missing something, but the epsilon class GSTs don't seem to have obviously higher expression in the midgut than the tubules or or integument.

Version 1:

Reviewer comments:

Reviewer #1

(Remarks to the Author)

The authors have thoroughly addressed all of my previous concerns by providing additional experimental data and making comprehensive revisions to the manuscript. I find the current version to be scientifically sound and suitable for publication in its present form.

Reviewer #2

(Remarks to the Author)

Re-review of the revised version.

I fully agree with the author's response to my comments and their modifications to the manuscript. I support publication in the present form.

I have a few very minor comments that I suggest the authors can follow or not according to their personal judgement.

Last comments to GST manus

Line 45-46: are ITCs usually the dominant products of GSLs? I simply don't know. In many vegetables, they are not. If you are also in doubt, you might want to change to: The classical hydrolysis products / the archetypic products or similar. But this is up to you.

Line 58 The lipophilicity: I would write: Lipophilicity. In this way you don't suggest that all ITCs are lipophilic.

Line 382: I guess as scientists we are supposed to test our hypotheses, aiming at disproving them. Hence, in line 382 I would change "prove" to "test".

References 7-8: I questioned the reference to Windsor et al. (2005) in the first version. However, if the alternatives are the present refs 7+8, I'd prefer Windsor & co. At least Windsor et al. used high confidence peak identification, and you can argue that the paper was pioneering in its approach (combination of GSL screen and phylogeny). I am worried about the relaxed attitude of Fahey et al and Edger et al. to peak identification (Edger et al. relied mainly on Fahey et al). I would not use those two in this context.

Better than Fahey et al. would be Daxenbichler et al. 1991 (Phytochemistry 30, 2623) or any newer high-quality screens or relevant critical reviews if there are any good. Fahey et al. mainly relied on the Daxenbichler screen, which is surprisingly reliable even after 35 years, but the review was messed it up by including all other reports with no attention to reliability.

Journal abbreviations: I think "one-word journal names" are not usually abbreviated. If this is true even for the present journal, "Phytochem." should be Phytochemistry throughout in the reference list.

Reviewer #3

(Remarks to the Author)

I appreciate the authors' effort to comprehensively address reviewer concerns. The revisions were thoughtful and effective. I especially appreciate the additional experiment to quantify the metabolic fates of ingested ITCs, and the effort to enhance reproducibility and reuse by sharing the data used to create each figure and table. I have some very minor final revisions to suggest below, and I would be comfortable that the authors could make these revisions without needing my inspection in a further round of review.

Overall, this study appears to have been carefully conducted, clearly presented, and represents an immense amount of effort. It will be a valuable contribution to the literature on the genetics, biochemistry, and physiology of how insect herbivores overcome blends of plant defensive compounds!

Fig S5a: It appears that not all possible comparisons between compounds were made. (For example, at a glance, I would assume allyl ITC is likely significantly different from more compounds than are indicated). Thus, I'm not sure which comparisons were made but aren't significant, vs. which were not made. It would be helpful to explain how comparisons were chosen, and to put "n.s." for the non-significant ones if not all possible comparisons were made.

L242-251: I am not an expert in these particular methods, but I felt that there was a lack of information about the motivation, methods (in a non-technical sense), and interpretation of the newly-added structural inference and molecular docking simulations. This should be elaborated somewhat in this portion of the results, and further in the discussion – so that this section can be understood by readers with different backgrounds, and so it truly feels like it's integrated into the paper and adds value.

- Andrew Gloss

Response to reviewers

Senior Editor (David Favero) and Editorial Board Member (Hannes Schuler):

We therefore invite you to revise and resubmit your manuscript, taking into account the points raised. In particular, please address all the objections regarding the interpretation and conclusion raised by Reviewer #2 and the comments regarding the presentation of the results by Reviewer #3. Also in accordance with Reviewer #3's comments, we strongly recommend that all source data are made available to the reviewers upon resubmission. Please also address all minor comments raised by the reviewers and ensure all changes made in response to the reviewers' comments are highlighted in the manuscript text file.

We are committed to providing a fair and constructive peer-review process. Do not hesitate to contact us if you wish to discuss the revision in more detail or if there are specific requests from the reviewers that you believe are technically impossible or unlikely to yield a meaningful outcome.

At the same time, we ask that you ensure your manuscript complies with our editorial policies.

Specifically:

For all graphs depicting a single point value (e.g., mean) with error bars, you must add individual data points or convert the graph to a boxplot or dot-plot to show data distribution.

It's mandatory to provide access to the numerical source data for graphs and charts either through a repository or by providing the data in a Supplementary Data file (in excel format).

All blots/gels must be accompanied by size markers in every figure panel. Uncropped and unedited blot/gel images must be included as Supplementary Figure(s) in the Supplementary Information pdf.

Please ensure that you have complied with the data deposition policies at the Nature Portfolio, please see here.

Please ensure that you have complied with our policies on research involving animals and humans, see here

Please follow the ARRIVE guidelines for reporting animal experiments. Please fully complete an ARRIVE checklist including both the essential and recommended set of items (adding information to the manuscript where needed) and upload this with your revised manuscript.

Please also see our revision checklist for guidance on formatting the manuscript and complying with our policies. A comprehensive guide to our formatting requirements for final submissions is also available for your reference here.

When submitting the revised version of your manuscript, please pay close attention to our Digital Image Integrity Guidelines.

Best regards,

Hannes Schuler and David Favero

I would like to thank you and the reviewers once again for the thoughtful and constructive comments on our manuscript. We greatly appreciate the opportunity to revise and enhance our work in response to these valuable suggestions.

Below, we provide a point-by-point response to all reviewer comments. Reviewer comments are presented in red, and our corresponding responses are highlighted in blue. All revisions made in response to the reviewers' feedback are clearly marked in the revised manuscript file.

In addition, we have submitted all raw data and included detailed "roadmaps" outlining the procedures used for data plotting and statistical analysis in the Supplementary Data files. We have also ensured that our manuscript complies fully with the journal's editorial policies. Specifically, all main figures now present individual data points with appropriate error bars. Furthermore, uncropped and unedited gel images, raw metabolomics data, and molecular docking input/output files have been deposited in the Edmond Open Access Data Repository (<https://doi.org/10.17617/3.WAOR7H>).

We sincerely hope that the revised version meets the standards for publication in *Communications Biology*.

Reviewer #1 (Remarks to the Author):

The authors present a comprehensive and well-executed study on the detoxification mechanisms of glucosinolate-derived isothiocyanates (ITCs) in *Spodoptera littoralis*, with a particular focus on the role of glutathione-S-transferases (GSTs). By integrating toxicological experiments, metabolomics, biochemical assays, and gene expression analysis, the study provides a holistic view of ITC detoxification in *S. littoralis*. The production of total 34 recombinant GSTs and the subsequent enzymatic assays effectively demonstrate the capacity of multiple GSTs to catalyze ITC-GSH conjugation, highlighting both the redundancy and broad substrate specificity of these enzymes. Overall, this work is resource-intensive and could be accepted after minor revisions.

1. When larvae were fed artificial diets containing ITCs, could the authors estimate the overall percentage of ITCs that were converted? This data is important for understanding the efficiency of the detoxification process.

We thank the reviewer for this valuable suggestion. To evaluate the conversion of 4MSOB isothiocyanate (ITC) into its conjugates by *Spodoptera littoralis*, larvae ($n = 20$) were individually placed in 35 mL polystyrene solo cups and starved for 20 hours. Each larva was then provided ad libitum access to a cube (~1 cm³) of artificial bean diet supplemented with 1 μ mol of 4MSOB ITC per gram of diet. To estimate initial diet weight, five additional diet cubes were placed in empty cups as controls. After 24 hours of feeding, the remaining diet was collected from both experimental and control cups. Larvae were then maintained in their respective cups for an additional 3 hours to allow for excretion. Frass and larvae were subsequently collected separately. All samples (diet, frass, and larvae) were freeze-dried and weighed. Diet consumption was determined by subtracting the remaining diet weight from the average initial weight

of control cubes. Each sample was extracted with 1 mL of 60% methanol and analyzed via LC-MS/MS (Lines 622–631).

The extent of 4MSOB ITC conversion was assessed by quantifying both the unmodified compound and its conjugates in the consumed diet, larval tissues, and frass. Approximately 36% of the ingested 4MSOB ITC was recovered as conjugates in the frass, while 8% remained in conjugated form within the larval body. This information has now been added to the Results section (Lines 169-173) and the data presented in Supplementary Figure 2.

Figure 1 (Supplementary Figure 2 in manuscript) Quantitative conversion of 4MSOB ITC into its conjugates during *S. littoralis* feeding. Total amounts of 4MSOB ITC and its conjugates were measured in the consumed diet, *S. littoralis* larvae, and collected frass ($n = 20$). Bold numbers in graph represent the mean values of 4MSOB ITC and its conjugates. Detailed measurements are provided in the raw data file. Statistical differences among means (\pm SE) of 4MSOB ITC or its conjugates were assessed using one-way ANOVA followed by Tukey's HSD test. Different lowercase letters indicate statistically significant differences ($P < 0.05$).

2. The study lacks genetic validation (e.g., CRISPR knockout of GST gene clusters or overexpression in *Drosophila* model systems) to confirm the physiological role of specific GSTs in ITC detoxification. The authors should acknowledge this limitation and discuss how future genetic studies could further validate the functional importance of GSTs *in vivo*.

We thank the reviewer for this suggestion. In response, we have now added a paragraph to the Discussion section (Lines 382–396) addressing the benefits of CRISPR-mediated knockout of GST gene clusters and overexpression approaches in *Drosophila* model systems as well as the limitations due to functional redundancy. This paragraph also outlines the potential for future genetic studies to further dissect the functional roles of GSTs in *S. littoralis*.

3. The comprehensive GST enzymatic activity data against different ITCs is a valuable resource. Would it

be possible to employ AlphaFold3 or similar AI-based models to gain structural insights that correlate with these data? A brief discussion of this possibility would enhance the study.

We thank the reviewer for this valuable suggestion. Utilizing protein structural models from the AlphaFold Protein Structure Database (<https://alphafold.ebi.ac.uk/>), we conducted molecular docking analyses using AutoDock Tools (v1.5.6, Molecular Graphics Laboratory, Scripps Research Institute) and visualized the docking results with the PyMOL Molecular Graphics System (v3.1.4.1, Schrödinger, LLC). A detailed description of the docking methodology is provided in Lines 745–759. The results of these analyses, which illustrate the interactions between GST proteins and ITC ligands, have been incorporated into the manuscript (Lines 241–251) and are presented in Figure 3E and Table 2.

Minor Comments:

4. Figure 1A-B: There is no positive control in the bioassays (e.g., a known growth inhibitor such as azadirachtin) to provide a reference for the relative toxicity of these ITCs. Including such a control would strengthen the interpretation of the results.

We thank the reviewer for this helpful suggestion. To strengthen our conclusions, we performed an additional experiment using the insecticides flufenoxuron and diflubenzuron as positive controls¹. All compounds, including ITCs and insecticides, were applied at a concentration of 1 µmol/g fresh weight of artificial diet. As shown in Supplementary Figure 1A, the relative growth rates of larvae fed with 4MSOB ITC were significantly reduced, and this reduction was not significantly different to that observed in larvae fed with flufenoxuron or diflubenzuron. However, as illustrated in Supplementary Figure 1B, most *S. littoralis* larvae died within five days of feeding on diets supplemented with flufenoxuron or diflubenzuron, whereas larvae fed on diets containing allyl ITC or 4MSOB ITC survived. These results suggest that although ITCs negatively affect larval growth and development, they do not exert insecticidal effects comparable to synthetic insecticides, which cause substantial larval mortality. The results of this experiment are now described in the main text (Lines 123-124, and Lines 129-130), and the corresponding figures have been included as Supplementary Figure 2.

Figure 2 (Supplementary Figure 1 in manuscript). Inhibition of *Spodoptera littoralis* larval growth by isothiocyanate (ITC) and insecticides. (A) Larval growth of *S. littoralis* was negatively affected by the presence of insecticides in the artificial diets ($n = 15$), with flufenoxuron causing a significant reduction in growth. Relative growth rates ($\text{mg}\cdot\text{mg}^{-1}\cdot\text{day}^{-1}$) were calculated as the natural logarithm (Ln) of final mass minus initial mass, divided by the number of feeding days (4 days for all treatments). (B) Larval survivorship was significantly affected by insecticide exposure ($n = 15$). Abbreviations: 4MSOB ITC, 4-methylsulfinylbutyl isothiocyanate. Box plots show the interquartile range (25th to 75th percentile), with median values indicated by horizontal lines and whiskers representing data range and outliers. Significant differences among means ($\pm\text{SE}$) in panel A were determined by one-way ANOVA followed by Tukey's HSD test. Survival analysis in panel B was performed using Kaplan–Meier survival estimates. Different lowercase letters indicate statistically significant differences ($P < 0.05$).

5. Figure 1D: The manuscript should clarify whether the *myb28myb29xcyp79b2cyp79b3* mutant is a transheterozygote or a double null allele.

We thank the reviewer for highlighting this important point. The *myb28myb29xcyp79b2cyp79b3* mutant used in our study is a quadruple knockout line (i.e., a combination of two double null alleles). It was generated by manually crossing the *cyp79b2cyp79b3* double mutant with the *myb28myb29* double mutant, and subsequently confirmed by genotyping and glucosinolate profiling in the F4 generation². We have added this information to the Materials and Methods section (Line 466) and corrected the corresponding reference accordingly.

Reviewer #2 (Remarks to the Author):

Versatile GST based detoxification Sun et al.

This is an important, interesting and well carried out investigation of the complex, general biochemical ways a generalist insect detoxifies a typical complex array of isothiocyanates from crops (such as mustards, canola and cabbages) and wild plants. It seems to be carried out with an open mind for complexity, and at relevant concentrations for the typical plant levels, as opposed to some previous studies carried out at

somewhat higher concentrations. I have few major objections to design, description, discussion and interpretation of the experiments, except a method concern, a discussion concern, and a concern that the final conclusion is a bit too strong.

1. Conclusion-concern: From six relatively equal sized ITCs, with no replication of the types of ITCs (only one unsaturated, only one SO-containing, modest size difference), I do not agree that one can conclude (line 397-399) “*S. littoralis* GSTs generally have less activity with ITCs having larger, more complex side chains as those with smaller, less complex side chains”. This statement is bound to be copied and cited without proper caution. I suggest to reword, e.g. “from the small diversity of ITCs tested here, *S. littoralis* GSTs would seem to have less activity with ITCs having”. Also same section line 393-394, I would write: “*S. littoralis* has a number of properties that seems to be appropriate for a generalist...” (seems to be instead of are). I tend to agree, but who are we to judge?

We thank the reviewer for the helpful modifications. You are absolutely right that it is not appropriate to draw broad conclusions based solely on the six ITCs tested, which are relatively similar in size. The revised statements more accurately reflect the limitations of our study and are more appropriately worded. Specifically, we have adopted the following revised sentences: “In conclusion, the GSL detoxification system of *S. littoralis* has a number of properties that seems to be appropriate for a generalist-feeding insect herbivore” (Lines 447-448), and “Nevertheless, from the small diversity of ITCs tested here, *S. littoralis* GSTs would seem to have less activity with ITCs having larger, more complex side chains as those with smaller, simpler side chains” (Lines 451–454).

2. The method concern is this: You calculate GST activity from absorbance change at 274 nm, using extinction coefficients of the substrates. But what is the extinction coefficient of the GST-ITC product at this wavelength. If the product has significant absorption, the initial rates would be under-estimated, it seems. Please could you comment on this in the revised version and adjust calculations if needed?

We thank the reviewer for bringing up this important point. We apologize for the lack of clarity in the original Materials and Methods section. The extinction coefficients used in the manuscript refer specifically to the ITC-GSH conjugates, rather than to the free ITCs themselves. Absorbance measurements were taken at 274 nm, which corresponds to the absorbance of the ITC–GSH conjugates. Prior studies have shown that the contribution of free ITCs to absorbance at this wavelength is negligible³.

⁴. We have now revised the Materials and Methods section to clearly indicate that the extinction coefficients refer to ITC–GSH conjugates (Lines 733-736). Additionally, we have included the absorbance spectra of both 4MSOB ITC and its GSH conjugate for reference at the range of 200-500 nm wavelength (Figure 3).

Figure 3. Absorbance spectra of 4MSOB ITC and 4MSOB ITC–GSH conjugate measured across the wavelength range of 200–500 nm. The dashed line indicates the absorbance at 274 nm.

3. The major discussion concern is this: Line 251-252: “isothiocyanates have been found to be the major toxic hydrolysis product derived from aliphatic GSLs” with a general ITC review as reference. But there are five major kinds of hydrolysis products from aliphatic GSLs: nitriles, ITCs, OZTs, epithionitriles and thiocyanates, as Jena-researchers and others have so beautifully described over the years. Do you trust this very general review for stating that ITCs are the major toxic products? Here, a broader view and some context would be appropriate, to make sure that the outsider don’t leave your paper with the impression that ITCs are the essence of GLs. In this paper, you have studied how a generalist copes with just one of the many weapons crucifers use for defending themselves. There is much to do still before we understand how generalists cope with the GL-MYR system.

You are absolutely right. We should not have referred only to ITCs in this context without mentioning other metabolites produced by the glucosinolate–myrosinase system. Accordingly, we have reworded the relevant sentences as follows:

“This diversity is amplified by the formation of different kinds of hydrolysis products from a single GSL, including ITCs, nitriles, oxazolidine-2-thiones and epithithioalkenes ⁵.” (Lines 281-283)

“Although we focused specifically on how *S. littoralis* copes with ITCs, the toxicity of other products generated by the glucosinolate–myrosinase system, as well as the counteradaptation strategies employed by generalist herbivores, warrant further investigation.” (Lines 297-300)

4. The abstract and title I agree with, neither exaggerate the findings, and the major concerns mentioned

above do not relate to these parts. But could you consider whether a revised title could include that a very large number of GSTs seemed to be involved, as I find this to be a significant finding.

We thank the reviewer for this excellent suggestion. In response, we have revised the title of our manuscript to: "Multiple glutathione-S-transferases detoxify diverse glucosinolate-based defenses in a generalist lepidopteran herbivore".

Minor:

In contrast, I have numerous minor points. In all cases, I would trust the authors to be the final authority as to follow my advice or not, but I ask the authors to consider the following points.

5. Hyphens or not. Since isothiocyanates are spelled in two words in chemistry (allyl isothiocyanate), I object to using hyphens in the abbreviations. It should be allyl ITC. Not allyl-ITC. This requires modification of all text and all figures.

We thank the reviewer for pointing this out. We have removed all unnecessary hyphens throughout the text and in all figures.

6. Gene abbreviations. The authors seem to consistently spell genes with standard font. But don't we usually abbreviate genes with italics and capitals (CYP79A2, GSTE2), and only use normal font for genotypes lacking that gene? If you agree, the knock in plants (e.g. line 179) and all reference to GST genes (e.g. line 238, line 387-388) should be capitalized. If you agree, this is a problem throughout the text in numerous places.

We thank the reviewer for pointing this out. We fully agree, and have modified the gene names to be in italics and uppercase where appropriate, both in the main text and in all figures.

7. Km or KM: I believe capital M is correct. The author who made Table 1 seems to agree. So I think you should capitalize M line 216.

Thank you for pointing this out. We have revised the text to use K_M instead of K_m , in accordance with standard biochemical nomenclature.

8. Testing of His-tagged proteins: Please make it crystal clear in abstract and everywhere in text, that the GSTs you tested for enzyme activity were not the native GSTs but His-tagged derivatives. His tags can be removed, but you like most other authors assumed it not needed. Future scientists may disagree, we can at least be very specific in pointing out whether derivatives or native enzymes are tested. Please do this consistently and in abstract, not only in methods.

We thank the reviewer for pointing this out. To clarify that the recombinant GST proteins used in this study are His-tagged derivatives, we have added this information to the Abstract, Introduction, Results, Discussion, and Methods sections.

9. Precise use of references. In the introduction, I several times (but not always) felt that references were used in a way not entirely loyal to the primary conclusions of those references. I also wondered whether some “Jenacentrism” had unconsciously been involved in selecting references. Please consider the following cases and reword or find other references as appropriate, if you agree with my concerns:

We apologize for relying primarily on references we were most familiar with. We sincerely thank the reviewer for encouraging us to broaden our perspective and incorporate relevant literature from outside our immediate field, which has helped to improve the quality and depth of our manuscript.

10. Line 30: I am aware that strict, careful counting has limited the number of concluded GLs to around that number, but do you agree that solid evidence for the existence of many more has been provided? Even if those additional are not 100% confirmed to structure, I think we can agree that there are many more, yet to be elucidated, than the current count. Can you in some way include this (e.g. “with c. 100 currently elucidated structures” or similar)?

We have revised the sentence as follows: “Among these, glucosinolates (GSLs), a group of amino acid-derived natural products with approximately 100 currently elucidated structures, serve as key defensive chemicals in Brassicales plants” (Lines 34–36).

11. Line 34: ref 8: is this really a new, state of the art reference for evolutionary diversification of GLs? (or is it the one that first comes to mind a Jena-based scientists?)

We apologize for initially citing the first reference that came to mind. We sincerely thank the reviewer for encouraging a more thorough review of the literature. In response, we have replaced reference 8 (Windsor et al., 2005⁶) with a more appropriate and up-to-date citation: Edger et al., 2015⁷.

12. Line 35, ref 9: Does the cited ref really back up this statement? To me, ref 9 seems to deal with the opposite problem, how plants with different GL profile influence the associated fauna...

Thank you for pointing this out. We have now replaced reference 9 (Newton et al., 2009⁸) with a more relevant and recent citation: Abuyusuf et al., 2023⁹.

13. Line 45: ref 10, do you find this to be a primary, authoritative reference on this specific topic? Ref 12: this seems to deal with using ITCs to cure cancer, quite something else than preventing cancer... Ref 13: ok.

Thank you for pointing this out. We have now replaced reference 10 (Rask et al., 2000¹⁰) with Bones et al., 1996¹¹ (Line 45) and removed the citation from Line 50. Additionally, reference 12 (Wu et al., 2009¹²) has been replaced with Zhang et al., 1994¹³ (Line 50).

14. Line 47-51: You write “Most research on the biological effect of ITCs has focused on those present in common crucifers, but has not looked at a variety of chemical structures”, and cite a review and a Jena paper that indeed only looked at a few. Although it is true that most papers test only one or a few ITCs, there are many excellent papers systematically testing a large number of GL products (GL+myrosinase) at conditions favoring ITCs. You actually cite one of them later (ref 46). I suggest to rephrase this section and provide the reader with the information that we have studies much larger than the present systematically testing structure-activity relationships (most based in the Bologna GL tradition). This concern also goes for the next sentence. You cite three papers (19-21, two from Jena) for our lack of ability yet to make generalizations. But what if you consider the Bologna papers in addition?

We sincerely thank the reviewer for pointing this out. We have reworded the sentence to improve clarity and accuracy, as follows: “In the plant order Brassicales, ITCs exhibit a wide variety of side-chain structures, which contribute to their diverse biological activities¹⁴” (Lines 57–58).

15. Line 51. Natural ITCs are not “typically lipophilic”. But sure, you decided to test mainly lipophilic ones. Please reconsider wording.

Thank you to the reviewer for pointing this out. We have now removed the word "typically" from Line 58.

16. Line 52. The inside of the cell is not an environment (environment = surroundings of the organism, right?). Intracellular space?

We have replaced the word "environment" with "space" in Line 60.

17. Line 54-55. This statement seems to prepare the reader with a possible explanation of higher activity of ITCs with a sulphinyl group. But you never catch up on this angle in the discussion. And do you really think that a sulphinyl could influence the ITC reactivity over a distance of four methylene groups? I'd suggest to delete this sentence and reference.

We have deleted the sentence and reference from Lines 61-62.

18. Line 55-56: This is wrong in my opinion, structure activity relationships have been investigated in detail in several systems (see above concerning line 47-51).

Thank you to the reviewer for pointing this out. We have now removed the phrase "but this has not yet been investigated in detail" from Line 64.

19. Line 684: If $P = 0.05$, your key will both result in n.s. and *, please revise.

Thank you to the reviewer for pointing this out. We have corrected the notation "n.s., $P \geq 0.05$; *, $P < 0.05$; **, $P < 0.01$; ***, $P < 0.001$ " throughout the manuscript and figures.

Figures

20. All. The final line in all legends, explaining the meaning of lowercase letters, need re-writing. Of course I can guess what you mean, but you really don't explain how to interpret the lowercase letters (I guess results NOT having the same lowercase letter but only different lowercase letters, are significantly different).

Thank you to the reviewer for pointing this out. We have corrected the statistical notation in all figure legends to: "Different lowercase letters in the graph denote statistically significant differences based on $P < 0.05$."

21. Fig. 1A 4MSOB, the text is formally wrong, it is a methylsulfinylalkyl side chain, isn't it?

Thank you to the reviewer. We have updated the text in Figure 1A.

22. Fig. 1B. How can I understand which curve represents which ITC, when there are three curves with three colors but seven colors in the key? I guess the experiments fell into three groups, perhaps you could state that blue indicates this, this and this, red indicates that, and that, green indicates....

Thank you to the reviewer for pointing this out. We have corrected Figure 1B by separating the overlapped lines, as shown below. We hope this improves clarity.

Figure 4 (Figure 1B in manuscript). The survivorship of *S. littoralis* larvae was not significantly influenced by ITC feeding ($n = 15$). Survival analysis was determined using Kaplan-Meier survival estimates.

23. Fig. 1D I was missing a color indicating “other” I guess there were a bit of other GLs in the leaves?

Thank you to the reviewer for pointing this out. We have added the contents of the other GLs in Figure 1D. These include 4OHI3M GSL, 7MSOH GSL, 4MTB GSL, and 1MOI3M GSL. The detailed concentrations are provided in Supplementary Table 1.

24. Fig. 2A,D, E. It took me a while to understand how the list of metabolites in red represents specific dots (the text at dots is VERY small). If you can in some way make this more clear, it would be fine. But if not, a reader like me will eventually get it.

Thank you to the reviewer for pointing this out. We have reconstructed the figure to improve clarity. The updated figure highlights the identified fractions using bold black-circled outliers and connects compound names to their corresponding fractions ($m/z(+)_min$) using bold numbers—for example, as shown in the revised figure below.

Figure 5 (Figure 2A in manuscript). Volcano plot of extracted LC-MS/MS features from non-targeted Q-TOF (UHPLC-HRMS, positive mode) analyses of frass of *S. littoralis* fed on an artificial diet supplemented with 4MSOB ITC compared to frass from feeding on an artificial diet without 4MSOB ITC (control) ($n = 5$ for each treatment).

Reviewer #3 (Remarks to the Author):

The ability to detoxify compounds in the environment (especially those in high concentrations in the diet) is key to persistence by ecological generalists, such as crop pests with broad host ranges. This is a complex process that may involve a diversity of related compounds and a diversity of related enzymes. Sun et al. investigate this process in a generalist lepidopteran that can feed on plants producing a variety

of related glucosinolate compounds (precursors to toxic isothiocyanates), and which encodes dozens of enzymes potentially capable of catalyzing their detoxification (glutathione S-transferases). Their work gives novel and well-supported insight into this process, and how it differs from simple textbook cases of highly specific detoxification of individual compounds by individual enzymes with high specificity. It's important to appreciate the potential for less specific processes as well, and this paper makes an excellent case.

The authors find a number of intriguing patterns, such as a rough correlation between larval performance on various isothiocyanate-containing diets and overall GST activity against those isothiocyanates, supporting the overall conclusion that “the propensity of multiple GSTs to react with an individual isothiocyanate and the wide expression of GST-encoding genes across larval organs likely support this generalist herbivore’s ability to thrive on glucosinolate-defended Brassicales plants.” This is an appropriate conclusion backed by a tremendous amount of work: characterizing the impact of particular toxins on larval performance, the accumulation of these toxins and the exact compounds produced by their metabolism *in vivo*, and the (inducible) expression and activity and substrate specificity of the entire repertoire of GST enzymes potentially capable of catalyzing this metabolism.

Particularly impressive was the rigor of the methods employed in this paper. For example: going above and beyond to include standards in their chemical profiling, even when this meant synthesizing standards that weren't available commercially; or utilizing both artificial diets and *Arabidopsis* glucosinolate mutants to investigate the effect of particular classes of glucosinolates.

I enjoyed the paper and think it will be an excellent contribution to the field. I just have a few revisions to suggest, relatively minor overall. Importantly, care must be taken to ensure that all data and relevant code or “roadmaps” to the analysis are shared to make the figures reproducible (point #1).

1. Data availability: I couldn't find the original data that went into creating many of the main figures, and it was not straightforward to search through the supplementary tables for the data I hoped to find (which I did not always successfully find). For example, for Figure 1, I found supplementary tables of fold changes for each LC-MS/MS feature but not a table of all detected intensities for all features across replicated measurements. I did see a note that all data will be made available upon publication. As part of this, I would like to in the supplement see a list of the input data files used for each figure panel, as well as a general roadmap to how that data was plotted and statistically analysed to create that figure panel, to make the figures reproducible. This would be not only a helpful resource for future readers, but an exercise to ensure that all necessary data is included. Preferably, I would like to see this input list / roadmap in any revised submission, as well as offering access for reviewers to actually review additional raw data that hasn't been shared yet and will be housed on external repositories so that they can assess

completeness. I am open to other approaches but overall I do strongly feel it's important to make data more complete and easier to navigate, and to make the figures more easily reproducible.

Thank you for your valuable feedback. We appreciate your emphasis on reproducibility. In response to your request, we now include a supplementary table listing the input data files used for each figure panel. Additionally, we provide a concise roadmap detailing the data processing steps, statistical analyses, and methods employed to generate each figure. This will ensure transparency and facilitate reproducibility.

2. Figures (general): the text is quite small, and I had to zoom in a lot on my computer. But the captions were taking a lot of space, and this may not be true in the final manuscript (where each figure could take up a full page, perhaps?). Please think carefully about if the text needs to be larger.

Thank you for your feedback regarding figure readability. We have carefully reviewed the text size in our figures and adjust it to improve clarity

3. Fig. 1A: What is the duration of the experiment for RGR (how many days passed)? Was it the same for all ITCs? It would be helpful to note this clearly in the caption.

Thank you for pointing this out. We have now added the relevant information to the legend of Figure 1A, including: "Relative growth rates ($\text{mg}\cdot\text{mg}^{-1}\cdot\text{day}^{-1}$) were calculated as the natural log (Ln) of the final mass minus the natural log of the initial mass, divided by the number of feeding days (12 days for all treatments)." Additionally, we have included similar information in the legends of Figure 1E and Supplementary Figure 1A, where relative growth rates were calculated over 3 and 4 days of feeding, respectively.

4. Fig. 1B: I cannot see most of the colored lines that should be there, based on the legend showing 7 lines.

Thank you to the reviewer for pointing this out. I have corrected Figure 1B by separating the overlapped lines, as shown below. I hope this improves clarity.

Figure 4 (Figure 1B in manuscript). The survivorship of *S. littoralis* larvae was not significantly influenced by ITC feeding ($n = 15$). Survival analysis was determined using Kaplan-Meier survival estimates.

5. Fig. 1A,D,E: I can't tell which points some of these labels correspond to. Please use lines to indicate the relationship between words and points. Or perhaps plotting a different symbol for identified and unidentified/unlabeled points would be sufficient.

Thank you to the reviewer for pointing this out. We have now reconstructed the figure to improve clarity. The revised figure now highlights the identified fractions using bold black-circled outliers and connects compound names to their corresponding fractions ($m/z(+)_min$) using bold numbers—for example, as shown in the revised figure below.

Figure 5 (Figure 2A in manuscript). Volcano plot of extracted LC-MS/MS features from non-targeted Q-TOF (UHPLC-HRMS, positive mode) analyses of frass of *S. littoralis* fed on an artificial diet supplemented with 4MSOB ITC compared to frass from feeding on an artificial diet without 4MSOB ITC (control) ($n = 5$ for each treatment).

6. L158-159: “Most of the 4MSOB-ITC and its derivatives were excreted in the frass with less than 1% remaining in the hemolymph (Figure 2C).” Unless I’m missing something, this is not the right experiment/analysis to make such a conclusion, or otherwise very specific logic must be provided to account for how the ITC and its derivatives were proportionately tracked through the organism. The figure shows concentrations of each compound in each tissue/substance. But there is much more hemolymph than frass, and this doesn’t seem to be accounted for (e.g., what is the hemolymph mass and the total mass of frass produced?). So even with a very high concentration in the frass, a substantial amount (more than 1% of what’s ingested) could remain in the hemolymph or other tissues. Please be more precise with logic or adjust the statement to simply refer to concentrations.

Thank you to the reviewer for pointing this out. We have removed the original sentence, “Most of the 4MSOB-ITC and its derivatives were excreted in the frass with less than 1% remaining in the hemolymph (Figure 2C),” and replaced it with a more accurate and data-driven statement: “The total conversion of 4MSOB ITC into its conjugates by *S. littoralis* was assessed by quantifying the amounts of 4MSOB ITC and its conjugates in the consumed artificial diet, larval bodies, and excreted frass. Approximately 36% of the ingested 4MSOB ITC was converted into conjugates and excreted in the frass, while 8% of the conjugated forms remained within the larval body (Supplementary Figure 2).” (Lines 169–174)

This revision is based on the results of a complementary experiment we conducted, as suggested by Reviewer #1, to evaluate the metabolic conversion of 4MSOB ITC in *S. littoralis*.

7. L205: It's a bit hard to see the support for this. Perhaps these comparisons could be plotted in the supplement and evaluated with a paired-sample Wilcoxon signed-rank test (with each GST being a “sample”)?

Thank you to the reviewer for this valuable suggestion. We have incorporated this improvement into Supplementary Figure 5A, as shown below.

Figure 6 (Supplementary Figure 5A in manuscript). *S. littoralis* GST enzyme activities for the conjugation of GSH with various ITCs. The average activity of each GST enzyme was treated as a single data point, and the total number of data points for each ITC was 34. Significant differences between means (\pm SE) were determined by a paired-sample Wilcoxon signed-rank test in A. Asterisks (*: $P < 0.05$; **: $P < 0.01$; ***: $P < 0.001$) denote statistically significant differences.

8. L230: Maybe I'm missing something, but the epsilon class GSTs don't seem to have obviously higher expression in the midgut than the tubules or or integument.

Thank you to the reviewer for pointing this out. You are absolutely right. The GST genes of the epsilon and delta classes were expressed across all tissues, but their expression was weak in the hemolymph. Accordingly, we have revised the text in the Results section to: “The GST genes of the epsilon and delta class were expressed across all tissues but weakly in the hemolymph (Figure 4A,B)”. (Lines 259-261)

References

1. Clarke, B.S. & Jewess, P.J. The inhibition of chitin synthesis in *Spodoptera littoralis* larvae by flufenoxuron, teflubenzuron and diflubenzuron. *Pestic. Sci.* **28**, 377-388 (1990).
2. Sun, J.Y., Sønderby, I.E., Halkier, B.A., Jander, G. & de Vos, M. Non-volatile intact indole glucosinolates are host recognition cues for ovipositing *Plutella xylostella*. *J. Chem. Ecol.* **35**, 1427-1436 (2009).
3. Kolm, R.H., Danielson, U.H., Zhang, Y., Talalay, P. & Mannervik, B. Isothiocyanates as substrates for human glutathione transferases: structure-activity studies. *Biochem.* **311**, 453-459 (1995).
4. Gloss, A.D. *et al.* Evolution in an ancient detoxification pathway is coupled with a transition to herbivory in the Drosophilidae. *Mol. Biol. Evol.* **31**, 2441-2456 (2014).
5. Jeschke, V., Gershenzon, J. & Vassão, D.G. Insect detoxification of glucosinolates and their hydrolysis products, in *Advances in botanical research*, Vol. 80. (ed. Stanislav, K.) 199-245 (Academic Press, 2016).
6. Windsor, A.J. *et al.* Geographic and evolutionary diversification of glucosinolates among near relatives of *Arabidopsis thaliana* (Brassicaceae). *Phytochem.* **66**, 1321-1333 (2005).
7. Edger, P.P. *et al.* The butterfly plant arms-race escalated by gene and genome duplications. *Proc. Natl. Acad. Sci. U.S.A* **112**, 8362-8366 (2015).
8. Newton, E.L., Bullock, J.M. & Hodgson, D.J. Glucosinolate polymorphism in wild cabbage (*Brassica oleracea*) influences the structure of herbivore communities. *Oecologia* **160**, 63-76 (2009).
9. Abuyusuf, M. *et al.* Glucosinolates and biotic stress tolerance in Brassicaceae with emphasis on cabbage: a review. *Biochem. Genet.* **61**, 451-470 (2023).
10. Rask, L. *et al.* Myrosinase: gene family evolution and herbivore defense in Brassicaceae. *Plant Mol. Biol.* **42**, 93-114 (2000).
11. Bones, A.M. & Rossiter, J.T. The myrosinase-glucosinolate system, its organisation and biochemistry. *Physiol. Plant.* **97**, 194-208 (1996).
12. Wu, X., Zhou, Q.H. & Xu, K. Are isothiocyanates potential anti-cancer drugs? *Acta Pharmacol. Sin.* **30**, 501-512 (2009).
13. Zhang, Y. & Talalay, P. Anticarcinogenic activities of organic isothiocyanates: chemistry and mechanisms. *Cancer Res.* **54**, 1976s-1981s (1994).
14. Manici, L.M., Lazzeri, L. & Palmieri, S. In vitro fungitoxic activity of some glucosinolates and their enzyme-derived products toward plant pathogenic fungi. *J. Agric. Food Chem.* **45**, 2768-2773 (1997).

Response to referees

Senior Editor (David Favero) and Editorial Board Member (Hannes Schuler):

Your manuscript entitled "Multiple glutathione-S-transferases detoxify diverse glucosinolate-based defenses in a generalist lepidopteran herbivore" has now been seen again by our referees, whose comments appear below. In light of their advice I am delighted to say that we are happy, in principle, to publish a suitably revised version in *Communications Biology*.

We therefore invite you to revise your paper one last time to address the remaining concerns of our reviewers. Please incorporate the final, minor suggestions from Reviewers #2 and #3 into your revised manuscript.

At the same time we ask that you edit your manuscript to comply with our format requirements and to maximise the accessibility and therefore the impact of your work.

* Please see the attached document for editorial requests for the final version (.docx file). Please ensure a completed version of this file is uploaded as a Related Manuscript with your final submission.

* Please review our final submission file checklist to ensure all necessary files are present with your final submission and to avoid delays in accepting your manuscript. For your reference, a style and formatting guide is available here and includes all of our style requirements.

* An updated editorial policy checklist that verifies compliance with all required editorial policies must be completed and uploaded with the revised manuscript. All points on the policy checklist must be addressed; if needed, please revise your manuscript in response to these points. Please note that this form is a dynamic 'smart pdf' and must therefore be downloaded and completed in Adobe Reader.

<https://www.nature.com/documents/nr-editorial-policy-checklist.pdf>

We sincerely thank the editors and reviewers for their thoughtful and constructive comments on our manuscript, both during the initial and current rounds of review. We greatly appreciate these valuable suggestions, which have significantly improved the quality and clarity of our work.

Below, we provide a detailed point-by-point response to all reviewer comments. Reviewer comments are presented in red, and our corresponding responses are highlighted in blue. All changes made in response to the reviewers' feedback are clearly marked in the revised manuscript.

We hope that the revised version meets the standards for publication in *Communications Biology*.

REVIEWERS' COMMENTS:

Reviewer #1 (Remarks to the Author):

The authors have thoroughly addressed all of my previous concerns by providing additional experimental data and making comprehensive revisions to the manuscript. I find the current version to be scientifically

sound and suitable for publication in its present form.

Reviewer #2 (Remarks to the Author):

Re-review of the revised version.

I fully agree with the author's response to my comments and their modifications to the manuscript. I support publication in the present form.

I have a few very minor comments that I suggest the authors can follow or not according to their personal judgement.

Last comments to GST manus

1. Line 45-46: are ITCs usually the dominant products of GSLs? I simply don't know. In many vegetables, they are not. If you are also in doubt, you might want to change to: The classical hydrolysis products / the archetypic products or similar. But this is up to you.

We thank the reviewer for this suggestion. In response, we have revised the sentence at lines 43-44 to: "Often regarded as the stereotypical dominant hydrolysis products of glucosinolates, the ITCs are toxic to many herbivores and pathogens."

2. Line 58 The lipophilicity: I would write: Lipophilicity. In this way you don't suggest that all ITCs are lipophilic.

We thank the reviewer for this suggestion. The revision has been made at line 52 to "ITC side chain lipophilicity should facilitate diffusion through lipid bilayer membranes,".

3. Line 382: I guess as scientists we are supposed to test our hypotheses, aiming at disproving them. Hence, in line 382 I would change "prove" to "test".

We agree with the reviewer and have replaced "prove" with "test" at line 364.

4. References 7-8: I questioned the reference to Windsor et al. (2005) in the first version. However, if the alternatives are the present refs 7+8, I'd prefer Windsor & co. At least Windsor et al. used high confidence peak identification, and you can argue that the paper was pioneering in its approach (combination of GSL screen and phylogeny). I am worried about the relaxed attitude of Fahey et al and Edger et al. to peak identification (Edger et al. relied mainly on Fahey et al). I would not use those two in this context.

Better than Fahey et al. would be Daxenbichler et al. 1991 (Phytochemistry 30, 2623) or any newer high-quality screens or relevant critical reviews if there are any good. Fahey et al. mainly relied on the Daxenbichler screen, which is surprisingly reliable even after 35 years, but the review was messed it up by including all other reports with no attention to reliability.

We thank the reviewer for the valuable feedback. We agree that the studies by Fahey et al. and Edger et al. are less suitable in this context due to limitations in compound identification. Accordingly, we have replaced them with Windsor et al. (2005), which provides a more rigorous and pioneering approach through high-confidence glucosinolate profiling combined with phylogenetic analysis. Additionally, we now cite Daxenbichler et al. (1991) as a reliable foundational reference. We appreciate the reviewer's suggestion, which has strengthened the reference framework of our manuscript.

5. Journal abbreviations: I think “one-word journal names” are not usually abbreviated. If this is true even for the present journal, “Phytochem.” should be Phytochemistry throughout in the reference list.

We thank the reviewer for this suggestion. The change has been made in the reference section accordingly.

Reviewer #3 (Remarks to the Author):

I appreciate the authors' effort to comprehensively address reviewer concerns. The revisions were thoughtful and effective. I especially appreciate the additional experiment to quantify the metabolic fates of ingested ITCs, and the effort to enhance reproducibility and reuse by sharing the data used to create each figure and table. I have some very minor final revisions to suggest below, and I would be comfortable that the authors could make these revisions without needing my inspection in a further round of review.

Overall, this study appears to have been carefully conducted, clearly presented, and represents an immense amount of effort. It will be a valuable contribution to the literature on the genetics, biochemistry, and physiology of how insect herbivores overcome blends of plant defensive compounds!

6. Fig S5a: It appears that not all possible comparisons between compounds were made. (For example, at a glance, I would assume allyl ITC is likely significantly different from more compounds than are indicated). Thus, I'm not sure which comparisons were made but aren't significant, vs. which were not made. It would be helpful to explain how comparisons were chosen, and to put “n.s.” for the non-significant ones if not all possible comparisons were made.

We appreciate the reviewer's comment. In Fig. S5a (Supplementary Figure 7), we performed pairwise comparisons between all groups. However, to ensure clarity and transparency, we have now revised the figure legend to explicitly state that all pairwise comparisons were conducted, and we have clarified the meaning of the statistical annotations used. The updated legend now is written: "Significant differences between means (\pm SE) were determined using a paired-sample Wilcoxon signed-rank test for all pairwise comparisons in (A), with detailed results provided in Supplementary Data 8, and using one-way ANOVA followed by Tukey's HSD test in (B). In (A), asterisks indicate statistically significant differences between groups (n.s. $P \geq 0.05$; * $P < 0.05$; ** $P < 0.01$; *** $P < 0.001$). In (B), different lowercase letters denote statistically significant differences at $P < 0.05$).

7. L242-251: I am not an expert in these particular methods, but I felt that there was a lack of information about the motivation, methods (in a non-technical sense), and interpretation of the newly-added structural inference and molecular docking simulations. This should be elaborated somewhat in this portion of the results, and further in the discussion – so that this section can be understood by readers with different backgrounds, and so it truly feels like it's integrated into the paper and adds value.

Thank you for this helpful suggestion. In response, we have clarified the motivation for molecular docking simulations in lines 228–229. Additionally, we have expanded the Discussion section, lines 338-343, to better emphasize their relevance to our central hypotheses.